# An airway-to-brain sensory pathway mediates influenza-induced sickness

Na-Ryum Bin[1], Sara L. Prescott[1,3], Nao Horio[1], Yandan Wang[1], Isaac M. Chiu[2] & Stephen D. Liberles[1✉]

Pathogen infection causes a stereotyped state of sickness that involves neuronally orchestrated behavioural and physiological changes[1,2]. On infection, immune cells release a 'storm' of cytokines and other mediators, many of which are detected by neurons[3,4]; yet, the responding neural circuits and neuro–immune interaction mechanisms that evoke sickness behaviour during naturalistic infections remain unclear. Over-the-counter medications such as aspirin and ibuprofen are widely used to alleviate sickness and act by blocking prostaglandin E2 (PGE2) synthesis[5]. A leading model is that PGE2 crosses the blood–brain barrier and directly engages hypothalamic neurons[2]. Here, using genetic tools that broadly cover a peripheral sensory neuron atlas, we instead identified a small population of PGE2-detecting glossopharyngeal sensory neurons (petrosal GABRA1 neurons) that are essential for influenza-induced sickness behaviour in mice. Ablating petrosal GABRA1 neurons or targeted knockout of PGE2 receptor 3 (EP3) in these neurons eliminates influenza-induced decreases in food intake, water intake and mobility during early-stage infection and improves survival. Genetically guided anatomical mapping revealed that petrosal GABRA1 neurons project to mucosal regions of the nasopharynx with increased expression of cyclooxygenase-2 after infection, and also display a specific axonal targeting pattern in the brainstem. Together, these findings reveal a primary airway-to-brain sensory pathway that detects locally produced prostaglandins and mediates systemic sickness responses to respiratory virus infection.

Respiratory infections caused by influenza and other pathogens are leading causes of death and hospitalization worldwide[6], and the recent COVID-19 pandemic has broadly disrupted human society. Feeling sick can be profoundly debilitating, and most people are sick several times a year. The sensation of sickness represents a neural response to infection that may provide a highly coordinated and adaptive strategy to promote recovery[1,2]. Animals infected with various pathogens display common behavioural and physiological responses that can include fever, lethargy, loss of appetite, headache and pain, mood changes and decreased socialization, suggesting a common sickness state involving shared neural circuits[2,7]. In addition to common symptoms, other sickness responses are tailored to the site of infection; for example, some respiratory infections induce cough, congestion and bronchoconstriction, whereas some gut infections induce nausea, diarrhoea and vomiting. Infection-specific behavioural responses suggest multiple body–brain communication pathways for pathogen detection.

Several cytokines and immune mediators can induce sickness behaviour when administered in isolation, including interleukins, interferons, tumour necrosis factor (TNF) and eicosanoids[1,2,4,8]. These and other cytokines, as well as pathogen-derived factors such as lipopolysaccharide, bacterial toxins and formyl peptides, can activate an assortment of central and/or peripheral sensory neurons[4,9]. Together, these observations raise the possibility that there are many pathways for neuro–immune crosstalk, although it is unclear when or whether each of these pathways is engaged during naturalistic infections.

Chemicals such as salicylic acid from willow bark, aspirin and ibuprofen block biosynthesis of key infection-induced lipid mediators through inhibition of cyclooxygenase enzymes, and have provided a historically effective approach to manage sickness symptoms[5,10]. Prostaglandin E2 (PGE2) is a key cyclooxygenase-dependent metabolite that evokes sickness behaviour, and knockout of other enzymes downstream of cyclooxygenase in the PGE2 biosynthesis pathway also ameliorates sickness responses[11]. PGE2 is detected by a small subfamily of 4 G-protein-coupled receptors (EP1–EP4)[12], and some but not all pyrogen-induced sickness responses are thought to be mediated by the EP3 receptor[2]. The EP3 receptor is expressed in various brain regions, including the hypothalamus and circumventricular organs as well as peripheral neurons, immune cells and many other cell types[12,13]. Region-specific knockout of the EP3 receptor in the median preoptic nucleus (MnPO) of the hypothalamus diminished lipopolysaccharide-induced fever responses[14], with PGE2 receptors in other areas reportedly being relevant for effects on arousal and feeding[2]. These and other findings have led to several possible models in which (1) PGE2 can directly cross the blood–brain barrier owing to its hydrophobicity, (2) PGE2 can be detected or enter the brain at circumventricular organs, and/or (3) PGE2 can be synthesized in the

[1]Howard Hughes Medical Institute, Department of Cell Biology, Harvard Medical School, Boston, MA, USA. [2]Department of Immunology, Harvard Medical School, Boston, MA, USA. [3]Present address: Department of Biology, Massachusetts Institute of Technology, Cambridge, MA, USA. ✉e-mail: Stephen_Liberles@hms.harvard.edu

brain itself[2,15,16]. Central PGE2 could then activate a distributed neural network, with different receptive brain regions such as the MnPO evoking particular aspects of a characteristic sickness response[17].

The roles of prostaglandin receptors in peripheral neurons have remained unclear, and furthermore, we reasoned that exogenous application of chemically defined pyrogens may produce a systemic inflammatory response, whereas natural infections in different locations and of varying intensities may instead trigger local neuro–immune response pathways that have remained unexplored.

## Influenza causes sickness through EP3

We developed a mouse model to characterize the neuronal mechanisms underlying influenza-induced sickness behaviour. Mice were infected by intranasal administration of influenza A virus PR/8/34 (H1N1), and monitored for characteristic sickness responses over the subsequent 10–20 days. Influenza infection decreased food intake, water intake, locomotion, body weight and survival in wild-type mice. Higher titres of virus inoculum increased the extent of sickness behaviour, with the most severe phenotypes arising six-to-seven days after infection (Fig. 1a). Influenza infection increased PGE2 levels in both plasma and bronchoalveolar lavage fluid (BALF) over a similar time frame, as measured by enzyme-linked immunoassay (ELISA) (Extended Data Fig. 1b). PGE2 administration also acutely inhibited food intake (Extended Data Fig. 2a), and fibre photometry measurements showed decreased activity of hypothalamic neurons expressing agouti-related peptide (AGRP) (Extended Data Fig. 2c), consistent with a decreased motivation to eat, rather than solely a physical inability to eat. We also observed a hypothermic response to influenza infection (Extended Data Fig. 3a), consistent with previous observations in mice[18]. Administration of ibuprofen (1 mg ml$^{-1}$ in drinking water, ad libitum) or aspirin (20 mg kg$^{-1}$, daily intraperitoneal injection) to influenza-infected mice decreased plasma PGE2 levels, restored feeding, water intake and body weight, and promoted survival from infection (Fig. 1b and Extended Data Figs. 1b and 3b). Ibuprofen- and aspirin-treated mice retained low-level decreases in feeding, drinking and motility without elevated PGE2, raising the possibility that other neuro–immune communication pathways might also contribute to the behavioural responses. Yet, ibuprofen and aspirin substantially decreased influenza-induced sickness behaviour, consistent with a key role for cyclooxygenase-2 metabolites in neuro–immune crosstalk.

Roles for multiple prostaglandin receptors have been proposed in sickness behaviours[2,19]. We administered selective antagonists for each PGE2 receptor daily after influenza infection, and measured different aspects of sickness behaviour. The EP3 receptor antagonist DG-041 effectively blocked influenza-induced sickness in each parameter measured and also promoted survival (Fig. 1c), with an effect magnitude similar to that of ibuprofen and aspirin. By contrast, antagonism of the EP1, EP2 or EP4 receptors had no effect on any measured parameter. The EP3-selective agonist sulprostone had the opposite effect of inhibiting food intake (Extended Data Fig. 2b). Thus, mice display characteristic behavioural changes to influenza virus infection through the action of PGE2 on EP3 receptors.

## Glossopharyngeal neurons and sickness

The EP3 receptor is expressed in several classes of central and peripheral neurons. To determine the key site of EP3 receptor action in influenza-induced sickness, we obtained mice with an allele (*Ptger3$^{flox}$*) for Cre-dependent knockout of the *Ptger3* gene[14], which encodes the EP3 receptor. We then crossed *Ptger3$^{flox}$* mice with either *Nestin-cre* or *Advillin-cre$^{ER}$* mice, which target Cre recombinase to most central or peripheral neurons[20,21], respectively (Extended Data Fig. 4a), and measured influenza-induced sickness behaviours. *Advillin-cre$^{ER}$*; *Ptger3$^{flox}$* mice were treated with tamoxifen to induce Cre-mediated

recombination at least one week before virus administration; prior tamoxifen treatment of control mice did not affect the subsequent behavioural responses to influenza infection (Extended Data Fig. 4b). Influenza-induced decreases in feeding, drinking, movement, body weight and survival were attenuated in *Advillin-cre$^{ER}$*; *Ptger3$^{flox}$* mice (Fig. 2b), with effect magnitudes similar to ibuprofen treatment, but persisted in *Nestin-cre*; *Ptger3$^{flox}$* mice (Fig. 2a). Similar effects of *Ptger3* gene deletion on sickness behaviour were observed when lower, less lethal doses of influenza virus were used (Extended Data Fig. 5). Mice with attenuated sickness behaviour displayed a similar recovery time, as normal feeding, drinking and motility were observed by about two weeks after infection in all experimental groups. These observations indicated that influenza induces sickness responses through PGE2 action on peripheral sensory neurons.

The EP3 receptor is expressed in several classes of peripheral neurons marked in *Advillin-cre$^{ER}$* mice, including vagal sensory neurons, glossopharyngeal sensory neurons, spinal sensory neurons of the dorsal root ganglia, and potentially in other neuron types[20,22]. Glossopharyngeal and vagal sensory neurons were considered primary candidates for the detection of respiratory pathogens as they account for most of the innervation in the upper and lower airways[23]. In mice, the soma of vagal (nodose and jugular) and glossopharyngeal (petrosal) sensory neurons are fused into a large superganglion on each side of the body[23]. We injected an adeno-associated virus (AAV) with a constitutive *cre* allele (AAV-cre) bilaterally into both nodose–jugular–petrosal (NJP) ganglia of *Ptger3$^{flox}$* mice. Effective knockout of *Ptger3* in NJP ganglia was confirmed two weeks after AAV injection by RNA in situ hybridization (Extended Data Fig. 6). NJP ganglion-targeted *Ptger3* knockout caused a marked attenuation of influenza-induced sickness behaviour (Fig. 2c). These findings indicate that influenza induces behavioural changes through EP3 receptor expressed on vagal and/or glossopharyngeal sensory afferents.

*Ptger3* is expressed in a subset of vagal and glossopharyngeal sensory neurons, as revealed by RNA in situ hybridization (Extended Data Fig. 6b) and analysis of single-cell transcriptome data (Fig. 3a). Single-cell RNA sequencing approaches revealed dozens of molecularly distinct NJP sensory neurons[22,24,25]. The highest *Ptger3* expression was observed in 6 neuron clusters: J1, J2, J3, NP2, NP9 and NP26 (Fig. 3a), with J denoting jugular and NP denoting nodose–petrosal neurons. We crossed *Ptger3$^{flox}$* mice to *Piezo2-IRES-cre* mice, in which J1, J2, J3 and other NJP neurons are labelled, and *Phox2b-cre* mice, in which NP2, NP9, NP26 and other NJP neurons are labelled. Influenza-induced sickness behaviours were attenuated in *Phox2b-cre*; *Ptger3$^{flox}$* mice across a range of viral titres, but not in *Piezo2-IRES-cre*; *Ptger3$^{flox}$* mice (Fig. 3b and Extended Data Figs. 7 and 8a), suggesting a role for either NP2, NP9 or NP26 neurons. To distinguish these neuron types, we next crossed *Ptger3$^{flox}$* mice to *Pdyn-IRES-cre*, *Oxtr-IRES-cre* and *Gabra1-IRES-cre* mice, which display differential targeting of NP26, NP2 and NP9 neurons. *Gabra1-IRES-cre*; *Ptger3$^{flox}$* mice displayed a marked attenuation of influenza-induced sickness behaviour similar to ibuprofen treatment, whereas no effect was observed in either *Pdyn-IRES-cre*; *Ptger3$^{flox}$* or *Oxtr-IRES-cre*; *Ptger3$^{flox}$* mice (Fig. 3b and Extended Data Fig. 8a). *Gabra1-IRES-cre*; *Ptger3$^{flox}$* mice additionally displayed an attenuated hypothermia response to influenza infection (Extended Data Fig. 3a). We also tested a role for TRPV1 neurons since their deletion affects survival following bacterial lung infections that cause lethal pneumonia[26]. However, influenza-induced sickness behaviour was normal in *Trpv1-IRES-cre*; *Ptger3$^{flox}$* mice (Extended Data Fig. 8b,c), and furthermore, GABRA1 neurons are predominantly *Trpv1*-negative. Thus, deletion of the EP3 receptor in a small subset of *Gabra1*-expressing peripheral sensory neurons (NP9 neurons) has a wide-ranging effect on behavioural responses to influenza infection.

Next, we examined how manipulations of sensory neurons affected viral transcript levels and cytokine production. Influenza infection induced PGE2 to similar levels in plasma and BALF of *Gabra1-IRES-cre;*

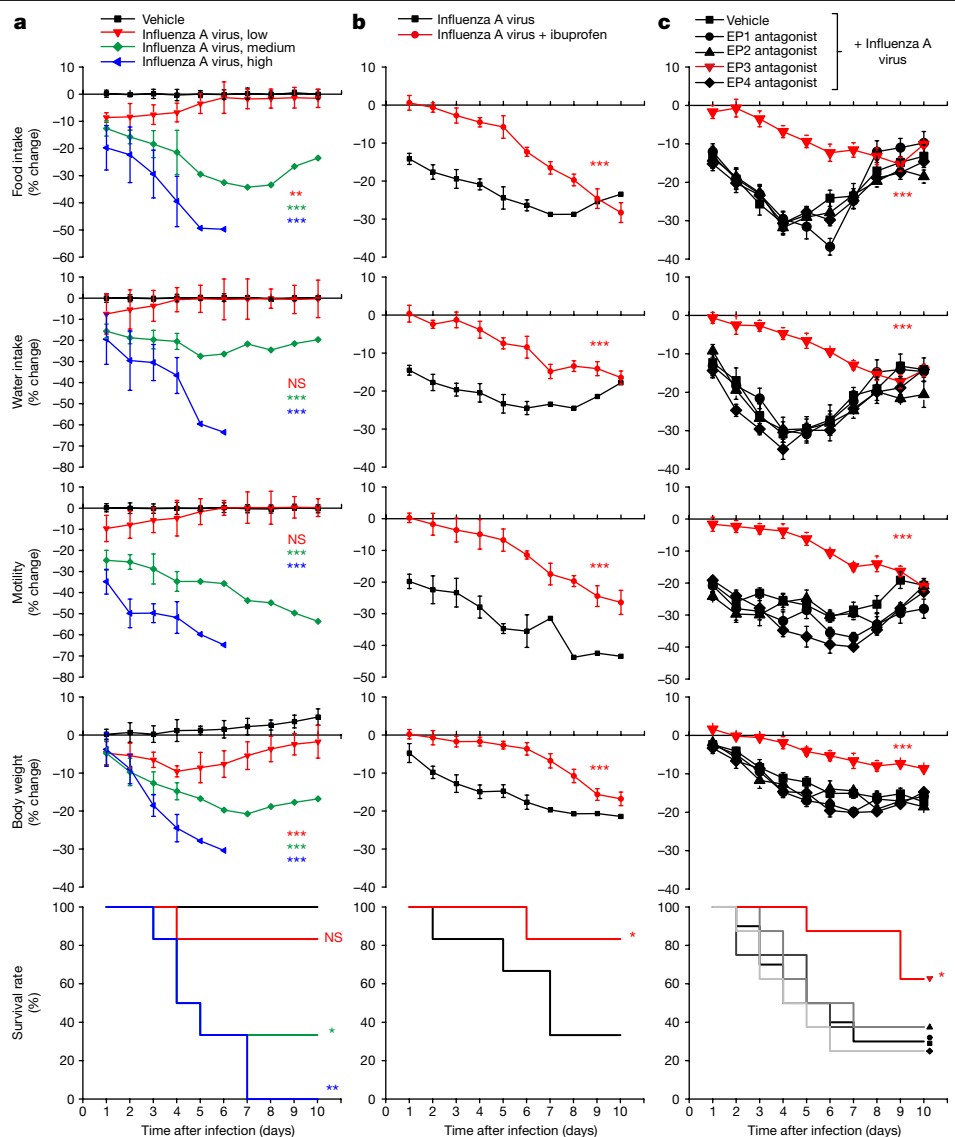

**Fig. 1 | Influenza infection induces multiple sickness symptoms through the EP3 receptor. a**, Mice were infected intranasally with 25 µl of influenza A virus at low ($10^5$ $EID_{50}$ $ml^{-1}$), medium ($10^6$ $EID_{50}$ $ml^{-1}$) or high ($10^7$ $EID_{50}$ $ml^{-1}$) dose and food intake, water intake, motility and body weight were subsequently monitored daily. $EID_{50}$ is the 50% egg infective dose. Data are mean ± s.e.m.; $n = 6$ mice per group. One-way ANOVA with Dunnett's multiple comparison test compared with vehicle control. Food intake: $P = 0.0048$ (low), $P < 0.0001$ (medium), $P < 0.0001$ (high). Water intake: $P = 0.8185$ (low), $P < 0.0001$ (medium), $P < 0.0001$ (high). Motility: $P = 0.5231$ (low), $P < 0.0001$ (medium), $P < 0.0001$ (high). Body weight: $P = 0.0002$ (low), $P < 0.0001$ (medium), $P < 0.0001$ (high). Survival: $P = 0.3173$ (low), $P = 0.0185$ (medium), $P = 0.0007$ (high). **b**, Mice were infected with influenza A virus (all infections are with $10^6$ $EID_{50}$ $ml^{-1}$ unless otherwise indicated), given drinking water with or without 1 mg $ml^{-1}$ ibuprofen, and monitored as indicated. Data are mean ± s.e.m.; $n = 6$ mice per group. Food intake: $P < 0.0001$; water intake: $P = 0.0001$; motility: $P = 0.0001$; body weight:

$P < 0.0001$; survival: $P = 0.0295$. **c**, Mice were infected with influenza A virus, injected daily (intraperitoneal injection, 1 mg $kg^{-1}$) with antagonists for EP1 (SC-51322), EP2 (PF-04418948), EP3 (DG-041) or EP4 (ONO-AE3-208), or vehicle alone, and monitored as indicated. Data are mean ± s.e.m.; $n = 10$ mice for vehicle groups; $n = 8$ mice for all others. For EP3 antagonist–food intake: $P < 0.0001$; water intake: $P < 0.0001$; motility: $P < 0.0001$; body weight: $P = 0.0002$; survival: $P = 0.0094$. **b**,**c**, Two-tailed unpaired $t$-test, with comparisons between groups treated with ibuprofen or EP3 antagonist and vehicle in **c**. Log-rank (Mantel–Cox) test for survival analyses; for behavioural or physiological changes, a mean daily change in behaviour (days 1–10 after infection or survival) was obtained for each mouse, and then used for comparisons across experimental groups (for more information see Extended Data Fig. 1a). *$P < 0.05$, **$P < 0.005$, ***$P < 0.0005$; NS, not significant.

*Ptger3^flox^, Phox2b-cre; Ptger3^flox^, Advillin-cre^ER^; Ptger3^flox^* and *Ptger3^flox^* mice (Extended Data Fig. 9a), consistent with changes in PGE2 detection rather than PGE2 synthesis underlying the observed behavioural differences. In control mice, viral transcript levels peaked three days after infection in the upper airways and five days after infection in the lungs. In *Gabra1-IRES-cre; Ptger3^flox^* mice, viral transcript levels were partially reduced in the upper airways and were decreased, delayed and persistent in the lungs (Extended Data Fig. 9b). Levels of interferon-gamma

(IFNγ), TNF and interleukin 6 (IL-6) similarly peaked in BALF of control mice 5 days after infection, and were likewise decreased and delayed in *Gabra1-IRES-cre; Ptger3^flox^* mice (Extended Data Fig. 9c). These findings indicate that targeted EP3 receptor knockout in sensory neurons not only affects sickness behaviour, but also the immune response and the transition from upper to lower respiratory tract infection.

We used a complementary approach involving diphtheria toxin-guided cell ablation to clarify a role for *Gabra1*-expressing

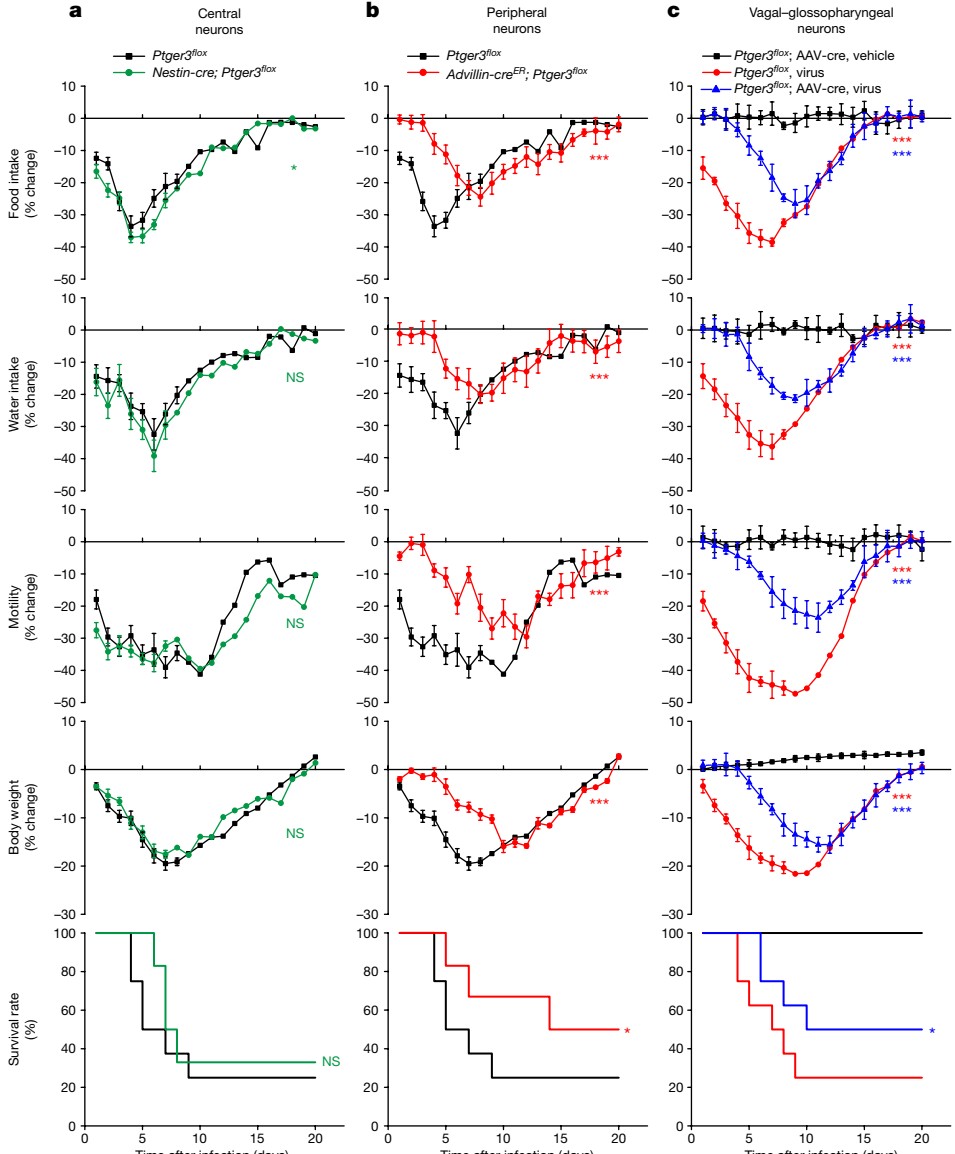

**Fig. 2 | Peripheral EP3 receptor is required for influenza-induced sickness.**
**a**,**b**, *Nestin-cre;Ptger3[flox]* (**a**), *Advillin-cre[ER];Ptger3[flox]* mice (**b**) or *Ptger3[flox]*
(**a**,**b**) mice were infected with influenza A virus and monitored as indicated.
Data are mean ± s.e.m.; *n* = 8 mice (*Ptger3[flox]*), *n* = 6 mice (*Nestin-cre* and
*Advillin-cre[ER]*). Two-tailed unpaired *t*-test as detailed in Fig. 1 for behaviour or
physiology analyses; log-rank (Mantel–Cox) test for survival analysis. **a**, Food
intake: *P* = 0.0126; water intake: *P* = 0.1006; motility: *P* = 0.0701; body weight:
*P* = 0.9361; survival: *P* = 0.7735. **b**, Food intake: *P* = 0.0004; water intake:
*P* = 0.0004; motility: *P* < 0.0001; body weight: *P* = 0.0004; survival: *P* = 0.0216.
**c**, The NJP ganglia of *Ptger3[flox]* mice were injected bilaterally with AAV-cre,
exposed to influenza A virus or saline and monitored as indicated. Data are
mean ± s.e.m.; *n* = 8 mice per group. *Ptger3[flox]*, virus−food intake: *P* < 0.0001;
water intake: *P* < 0.0001; motility: *P* < 0.0001; body weight: *P* < 0.0001;
survival: *P* < 0.0001. *Ptger3[flox]*; AAV-cre, virus−Food intake: *P* < 0.0001; water
intake: *P* < 0.0001; motility: *P* < 0.0001; body weight: *P* < 0.0001; survival:
*P* = 0.0376. One-way ANOVA with Dunnett's multiple comparison test as
detailed in Fig. 1 for behaviour or physiology analyses; log-rank (Mantel–Cox)
test for survival analysis, with comparisons made between *Ptger3[flox]*, virus and
*Ptger3[flox]*; AAV-cre, vehicle (red stars) or between *Ptger3[flox]*, virus and *Ptger3[flox]*;
AAV-cre, virus (blue stars).

vagal–glossopharyngeal sensory neurons in influenza-induced sickness, and rule out a role for other *Gabra1* expression sites. Mouse cells are normally resistant to diphtheria toxin, but can be rendered sensitive by the expression of the human diphtheria toxin receptor (DTR). NJP ganglia of *Gabra1-IRES-cre; lsl-DTR* mice were bilaterally injected with diphtheria toxin (we term these *Gabra1*-ABLATE mice), resulting in highly efficient ablation of vagal and glossopharyngeal GABRA1 neurons (Fig. 4c). We previously demonstrated that this approach does not affect Cre-negative neurons in NJP ganglia, or remotely located Cre-positive neurons[27]. *Gabra1*-ABLATE mice displayed attenuated sickness responses to influenza infection that were similar to those in *Gabra1-IRES-cre;Ptger3[flox]* mice or ibuprofen-treated mice (Fig. 4a).

NP9 neurons lack expression of *Hoxb4*(ref. [22]), and GABRA1 neurons are often clustered within NJP ganglia near the glossopharyngeal branch (Extended Data Fig. 10a), suggesting they comprise part of the glossopharyngeal nerve rather than the vagus nerve. Since diphtheria toxin injection similarly affected vagal and glossopharyngeal neurons, we distinguished the contributions from these nerves by transecting the glossopharyngeal nerve bilaterally while preserving the vagus nerve. Mice with bilateral glossopharyngeal nerve transection displayed a similar attenuation of influenza-induced sickness (Fig. 4b). PGE2 was previously shown to activate and/or induce transcriptional changes in spinal and vagal afferents[28–31], so we tested whether GABRA1 neurons are also directly activated. We observed that PGE2 directly evoked

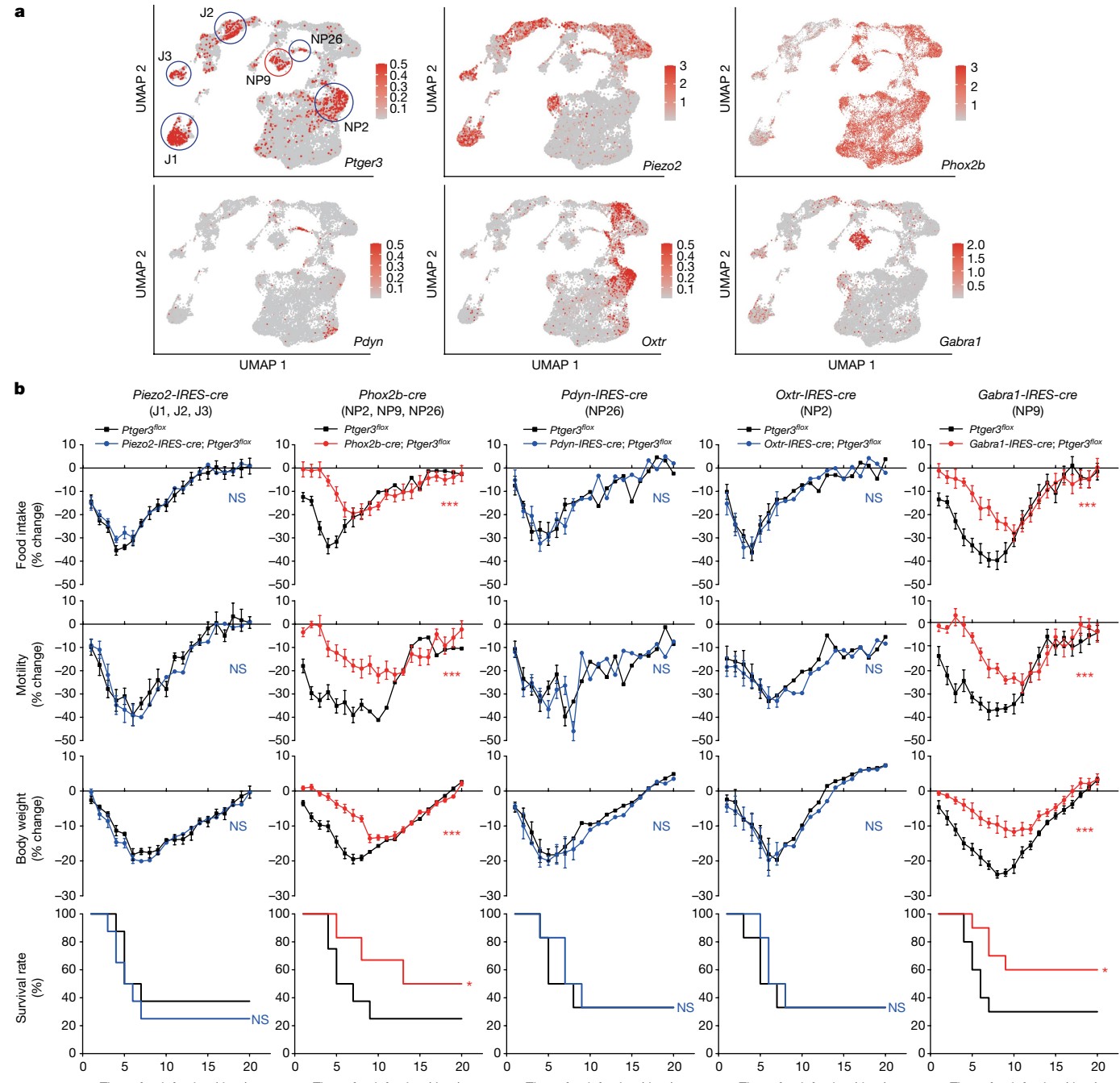

**Fig. 3 | Rare transcriptome-defined sensory neurons mediate influenza responses. a**, A uniform manifold approximation and projection (UMAP) plot derived from published single-cell transcriptome data of vagal and glossopharyngeal sensory ganglia[22] showing expression of indicated genes (colour shows relative expression on a natural log scale). **b**, Cell types, as highlighted in **a**, express the indicated gene as well as *Ptger3*. Mice were infected with influenza A virus and monitored as indicated. Data are mean ± s.e.m.; $n = 8$ (*Piezo2-IRES-cre*), $n = 6$–8 (*Phox2b-cre*; 8 control and 6 *Phox2b-cre*), $n = 6$ (*Pdyn-IRES-cre*), $n = 6$ (*Oxtr-IRES-cre*) and $n = 10$ (*Gabra1-IRES-cre*) mice per group. Two-tailed unpaired *t*-test as detailed in Fig. 1 for behaviour or physiology analyses; log-rank (Mantel–Cox) test for survival analysis. *Piezo2-IRES-cre*–food intake: $P = 0.2631$; motility: $P = 0.2680$; body weight: $P = 0.7925$; survival: $P = 0.4736$. *Phox2b-cre*–food intake: $P < 0.0001$; motility: $P < 0.0001$; body weight: $P = 0.0002$; survival: $P = 0.0181$. *Pdyn-IRES-cre*–food intake: $P = 0.4539$; motility: $P = 0.4946$; body weight: $P = 0.2675$; survival: $P = 0.7937$; *Oxtr-IRES-cre*–food intake: $P = 0.4786$; motility: $P = 0.6333$; body weight: $P = 0.3297$; survival: $P = 0.7121$; *Gabra1-IRES-cre*–food intake: $P = 0.0001$; motility: $P < 0.0001$; body weight: $P = 0.0004$; survival: $P = 0.0263$.

calcium transients in GABRA1 neurons acutely cultured from NJP sensory ganglia of *Gabra1-IRES-cre; lsl-tdTomato* mice (Fig. 4d, 16 out of 26 or 61.5% of tdTomato-positive, KCl-responsive neurons). Thus, rare glossopharyngeal NP9 neurons detect the presence of a respiratory viral infection indirectly through the EP3 receptor and, in response, evoke a multi-pronged programme of sickness behaviour.

## A sensory arc from nasopharynx to brain

*Gabra1-IRES-cre* mice provide a valuable tool for visualizing scarce glossopharyngeal sensory neurons involved in neuro–immune cross-talk, so we used genetic mapping approaches to mark the peripheral and central projections of GABRA1 neurons (Fig. 5a). NJP ganglia of

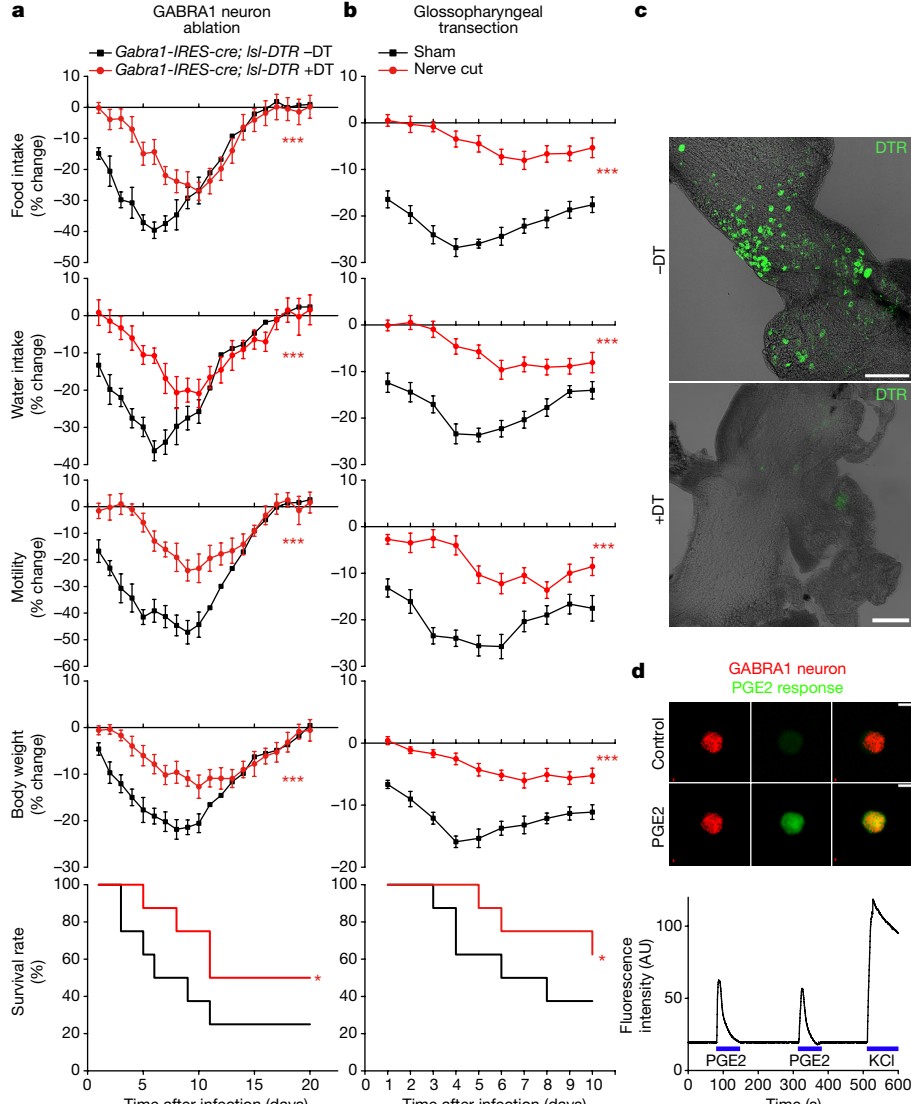

**Fig. 4 | PGE2 acts via glossopharyngeal sensory neurons. a**, *Gabra1-IRES-cre; lsl-DTR* mice were injected bilaterally in NJP ganglia with or without diphtheria toxin (DT) and then infected with influenza A virus and monitored as indicated. Data are mean ± s.e.m.; *n* = 8 mice per group. Food intake: *P* < 0.0001; water intake: *P* < 0.0001; motility: *P* < 0.0001; body weight: *P* = 0.0004; survival: *P* = 0.049. **b**, Wild-type mice with bilateral glossopharyngeal nerve transection surgery or sham surgery were infected with influenza A virus and monitored as indicated. Data are mean ± s.e.m.; *n* = 8 mice per group. Food intake: *P* < 0.0001; water intake: *P* < 0.0001; motility: *P* < 0.0001; body weight: *P* < 0.0001; *P* = 0.036.

Two-tailed unpaired *t*-test as detailed in Fig. 1 for behaviour or physiology analysis; log-rank (Mantel–Cox) test for survival analysis. **c**, Immunostaining for DTR in whole-mount preparations of NJP ganglia four weeks after injection. Scale bars, 200 μm. **d**, Calcium transients evoked by PGE2 (1 μM) or KCl (150 mM) were imaged using Calbryte 520 AM in tdTomato-positive neurons acutely collected from NJP ganglia of *Gabra1-IRES-cre; lsl-tdTomato* mice. Scale bar, 10 μm. Images are representative of three technical replicates. AU, arbitrary units.

*Gabra1-IRES-cre* mice were injected with either an AAV containing a Cre-dependent reporter gene encoding tdTomato (AAV-flex-tdTomato) or alkaline phosphatase (AAV-flex-AP), as well as an AAV containing a Cre-independent reporter gene encoding GFP (AAV-GFP) to visualize the global NJP projection field. Fibres were then visualized in the airways and the brain.

Centrally, sensory neurons of the nodose and petrosal ganglia cross the skull and innervate brainstem regions that include the nucleus of the solitary tract (NTS) and area postrema[32], whereas jugular sensory neurons innervate the paratrigeminal nucleus[33]. Various Cre-defined vagal afferents target spatially restricted subregions of the NTS, with some but not all afferent types also accessing the area postrema[24,34,35]. The axons of NJP GABRA1 neurons were highly restricted in the lateral NTS (Fig. 5a), not observed in the area postrema, and remote

from NTS locations targeted by axons of some other Cre-defined NJP neurons, such as gut-innervating neurons marked by expression of *Gpr65* or *Glp1r*[35]. These findings further support a model for topographic organization in the brainstem[36], with neurons relaying the presence of an airway infection spatially restricted from at least some other interoceptive inputs.

In the periphery, labelled GABRA1 axons were observed in only a few internal organs known to receive vagal and glossopharyngeal innervation. We observed densest innervation in the nasopharynx and oral cavity including taste papillae, some innervation of tracheal and pharyngeal muscles, and sparse innervation of stomach muscle (Extended Data Fig. 10b). Innervation was not observed in many other internal organs, including the heart, oesophagus, aorta and carotid sinus. The nasopharynx is a rich site for immune surveillance, as it

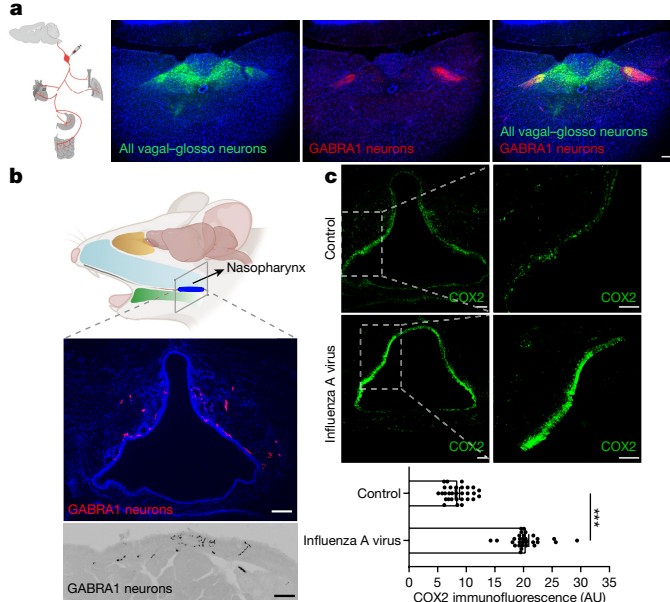

**Fig. 5 | GABRA1 neurons provide an airway–brain communication route.**
**a**, NJP ganglia of *Gabra1-IRES-cre* mice were injected bilaterally with Cre-independent AAV-GFP and Cre-dependent AAV-flex-tdTomato, and fluorescent axons were visualized (green: all NJP sensory axons; red: GABRA1 NJP axons) by immunohistochemistry for tdTomato and GFP in fixed coronal cryosections of mouse brainstem. Scale bar, 200 μm. **b**, NJP ganglia of *Gabra1-IRES-cre* mice were injected bilaterally with AAV-flex-tdTomato, and axons were visualized by immunostaining for tdTomato in fixed coronal cryosections of nasopharynx. Scale bars, 100 μm (top), 50 μm (bottom). **c**, Top, immunohistochemistry of COX2 in fixed cryosections of nasopharynx in uninfected mice (control) or mice infected for five days with influenza A virus. Scale bars, 100 μm. Bottom, quantification of COX2 immunofluorescence in the nasopharynx of indicated mice. Data are mean ± s.e.m.; *n* = 29 sections from 5 mice per group over 3 independent experiments. Two-tailed unpaired *t*-test, *P* < 0.0001. Left panel in part **a** adapted with permission from ref. [35], Elsevier. Top panel in part **b** created with BioRender.com.

connects the nasal cavity to the rest of the respiratory system and provides an early line of mucosal immune defence[37]. Influenza infection increases PGE2 levels locally within the nasopharynx[38], and in humans leads to inflammation of nasopharyngeal tonsils[39]. Mice have an orthologous system known as nasopharynx-associated lymphoid tissue[40], so we tested whether GABRA1 axons were located near relevant immune mediators. GABRA1 axons were enriched dorsally in epithelial and subepithelial layers of the nasopharynx (Fig. 5b) and notably, cyclooxygenase-2 expression was likewise detected in dorsal epithelium of the nasopharynx (Fig. 5c). Moreover, cyclooxygenase-2 expression was markedly upregulated in dorsal nasopharynx after influenza infection in mice (Fig. 5c). Thus, GABRA1 neurons occupy a strategic and privileged position for detection of the increased PGE2 levels that occur during viral infection.

## Discussion

Despite seasonal vaccination and antiviral therapeutics, influenza A viral infection remains one of the most severe human threats, affecting millions of people worldwide each year. Cyclooxygenase-2 inhibitors are among the most prevalent anti-sickness and anti-pain medications, with around 30 billion doses of nonsteroidal anti-inflammatory drugs (NSAIDs) consumed annually in the USA alone[41]. Blockade of PGE2 production alleviates sickness, and several models have been proposed for how the brain receives input about a peripheral infection through PGE2 (ref. [2]). Here, we observe that influenza-induced

sickness behaviours were similarly attenuated by 1) ibuprofen, aspirin and EP3 receptor antagonism, 2) *Ptger3*-targeted gene knockout using *Advillin-cre*[ER], *Phox2b-cre* and *Gabra1-IRES-cre* mice, and direct AAV-cre injection in NJP ganglia, 3) ablation of GABRA1 NJP neurons, and 4) and glossopharyngeal nerve transection. From these combined results, we conclude that a small cluster of *Gabra1*-expressing glossopharyngeal sensory neurons detects PGE2 and induces a neuronally orchestrated state of sickness associated with a suite of behavioural responses to respiratory viral infection.

Prostaglandins and many other cytokines can activate vagal and other peripheral afferents in vitro[8,29,31], but their physiological roles in naturalistic infections have been difficult to parse. Vagotomy blocks sickness responses to cytokines and lipopolysaccharide in some studies but not others[42,43], and this variability may be owing to the location or dose of pyrogen administration. Our findings indicate a clear role for sparse glossopharyngeal sensory neurons that are anatomically poised to detect influenza-induced PGE2. PGE2 has limited stability in vivo[44], so peripheral sensory neurons that directly detect PGE2 at the infection site would seemingly provide a fast and robust conduit for information transfer to the brain.

Sickness behaviours have been proposed to help conserve energy to fight off infection[1]. However, eliminating the influenza detection pathway of the glossopharyngeal nerve not only attenuates sickness behaviour, but also promotes survival. Knockout of EP3 from a small group of glossopharyngeal neurons is sufficient to mimic the profound survival phenotype observed in full-body knockouts. One model is that pathogen-induced anorexia is beneficial in some infection models but harmful in others. For example, blocking PGE2 production is harmful during mycobacterium 3 infection[45], but promotes survival during influenza infection[19,46]. Moreover, glucose gavage is detrimental during bacterial sepsis, where blood glucose may provide direct fuel for the pathogen, but protective during influenza infection[47]. Thus, disrupting neuronal responses to infection may decrease mortality by attenuating changes in feeding behaviour. Moreover, an infection-activated, anorexia-promoting neural pathway may provide a net survival benefit during certain bacterial infections, and thus is maintained over evolution, but is harmful during viral infection, as observed here. In addition, sickness behaviour may provide a separate population-level benefit as sick animals seek isolation and thereby limit pathogen transmission to kin[48].

These findings indicate that targeted EP3 receptor knockout in sensory neurons not only affects sickness behaviour, but also the immune response and viral transition from the upper to lower respiratory tract. A parsimonious interpretation of these findings is that petrosal GABRA1 neurons, on detection of PGE2, engage neural circuits that evoke coordinated responses that include sickness behaviours as well as motor reflexes that affect immune function. Additionally, changes in feeding behaviour may secondarily affect immune function[47,48] and activation of petrosal GABRA1 neurons may affect levels of cytokines other than PGE2 (Extended Data Fig. 9c), which can potentially elicit further neuronal feedback and enhance sickness behaviour. All such responses to infection would critically depend on the EP3 receptor in petrosal GABRA1 neurons, which serves an essential role in first relaying the presence of an upper respiratory infection to the brain, ultimately evoking sickness behaviour.

Influenza infection-induced sickness behaviour was attenuated, but not eliminated, following NSAID treatment, targeted EP3 receptor knockout, targeted neuronal ablation and glossopharyngeal nerve transection, suggesting other routes to sickness. Of note, the kinetics of residual sickness behaviour following manipulation of petrosal GABRA1 neurons matches the time course of virus accumulation in the lungs, indicating that there are probably two phases of influenza-induced sickness behaviour. The first phase occurs when the virus is most prevalent in the upper respiratory tract, and sickness is primarily mediated by PGE2-detecting glossopharyngeal sensory neurons marked by GABRA1,

which project to the nasopharynx. The second phase of sickness occurs when the virus is most prevalent in the lower respiratory tract, and the lung is primarily innervated by vagal rather than glossopharyngeal sensory neurons. Our data raise the possibility that this second phase of sickness involves another neuronal pathway independent of PGE2, EP3 and glossopharyngeal sensory neurons. Inhibitors of residual neuro–immune interaction pathways, once identified, could potentially provide improved relief of influenza-induced malaise when used in combination with NSAIDs. We also note that PGE2 receptors are expressed in multiple NJP sensory neuron types, as well as in the brain and dorsal root ganglia. It is likely that the nervous system uses multiple sensory pathways to detect peripheral infections, with different sensors perhaps specialized for particular pathogen types or infection locations in the body. In this way, gut and respiratory sensory neurons could potentially engage different neural circuits that lead, for example, to either cough or nausea[31,49]. Consistent with this notion, various routes to sickness are differentially sensitive to pharmacological inhibition; for example, gut malaise is treated clinically using antagonists of the serotonin receptor HTR3A[50]. Understanding the diversity of sensory pathways to sickness and when they are engaged by different pathogen infections will provide an essential framework for deciphering this complex and poorly understood physiological state, and may enable improved therapeutic interventions.

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

## Methods

### Animals

All animal procedures followed the ethical guidelines outlined in the National Institutes of Health Guide for the Care and Use of Laboratory Animals, and all protocols were approved by the institutional animal care and use committee (IACUC) at Harvard Medical School. Mice were maintained under constant temperature ($23 \pm 1$ °C) and relative humidity ($46 \pm 5\%$) with a 12-h light:dark cycle. Wild-type C57BL/6J (000664), *Nestin-cre* (003771), *Phox2b-cre* (016223), *Advillin-cre^ER* (032027), *Trpv1-IRES-cre* (017769), *Pdyn-IRES-cre* (027958), *Piezo2-EGFP-IRES-cre* (027719), *Oxtr-IRES-cre* (031303), *Agrp-cre* (012899), *lsl-DTR* (007900), *lsl-tdTomato* (Ai14, 007914) were purchased from Jackson Laboratories. *Gabra1-IRES-cre, lsl-L10-GFP* and *Ptger3^flox* mice were previously generated[14,22,51].

### Influenza A virus inoculation

Influenza A/PR/8/34 (H1N1) was purchased from Charles River Avian Vaccine Services (10100374) with a $EID_{50}$ of $10^{12}$ ml$^{-1}$. Virus was diluted in sterile saline for the administered dosage, which was $EID_{50}$ $10^6$ ml$^{-1}$ unless otherwise indicated. Viral administration was adapted from previous studies[52]. Awake mice were manually restrained and influenza A virus was delivered through each nostril (25 µl total volume) slowly using a P200 pipette to prevent spillage or delivery to the mouth. After administration, the mice were held with nostrils upright for 10–15 s, and returned to their cage.

### Behavioural monitoring

For analysis of influenza A virus infection-induced behavioural changes, six- to eight-week-old mice without sex bias were singly housed with free access to food and water. Baseline values were obtained by measuring daily food intake, daily water intake, body weight, and motility for three days before inoculation. After viral infection, all measurements were expressed as a percentage change from baseline values. Food and water intake were calculated daily by measuring the change in weight of remaining food and water in the cage. Animal movement was recorded (C922x Pro Stream Webcam, Logitech) in the home cage for 30 min daily and motility expressed as the total distance moved, as determined by the MTrack2 plug-in available in ImageJ (1.53q, NIH). For measurement of acute food intake, mice were fasted for 24 h, and at dark onset, were administered with PGE2 or sulprostone (EP3 agonist, 14765, Cayman Chemical Company) via the exposure route and doses indicated in Figure legends. Animals were then singly housed in clean cages with ad libitum access to food and water, and the amount of food consumed for 1 h determined by measuring the food in the cage before and after the test period.

### Body temperature measurements

Core body temperature was monitored by a rectal probe or radio telemetry. Rectal probe measurements were made hourly each day from noon to midnight using a mouse rectal temperature probe (Kent Scientific, RET-3, 0.16 mm tip diameter) connected to a thermometer (Kent Scientific WD-20250-9). Probes were sterilized (70% ethanol), lubricated (Aquagel lubricating gel, ParkerLabs, 57-05), and inserted ~5–7 mm into the rectum of physically restrained mice, and temperature was recorded for 30 s. For radio telemetry, a radio telemetry device that reports body temperature (HD-X11, DSI) was surgically implanted in the abdominal cavity. Surgery was performed under anaesthesia with avertin (200 mg kg$^{-1}$, intraperitoneal injection) and the abdominal area was disinfected and shaved. A -1 cm incision was made, and the abdominal peritoneum at the linea alba was gently exposed to insert the sterile radio telemetry device. The surgery site was sealed with VICRYL Sutures (J392H, Ethicon). Animals received Buprenorphine SR (BupSR-LAB, ZooPharm, 20 mg kg$^{-1}$ subcutaneously at the back of the neck), and were allowed to recover for at least a week. Body temperature data

were collected over 15 min intervals with a receiver (easyTEL, Emka Technologies) connected to a computer running iox software v.2.10.8. All data points from one mouse on a given day were averaged.

### Pharmacological manipulations

Ibuprofen (I4883) and aspirin (acetylsalicylic acid, A5376) were purchased from Sigma, and PGE2 receptor antagonists SC-51322 (2791), PF-04418948 (4818), DG-041 (6240), and ONO-AE3-208 (3565) were purchased from Tocris. Ibuprofen (1 mg ml$^{-1}$, saline) was added to drinking water ad libitum, and aspirin (20 mg kg$^{-1}$, saline) and PGE2 receptor antagonists (1 mg kg$^{-1}$, saline) were injected (intraperitoneally) daily. Administration of cyclooxygenase-2 inhibitors and PGE2 receptor antagonists began 3 days before virus inoculation and continued to the end of the monitoring period. Tamoxifen was administered daily for 5 days (intraperitoneal injection, 70 mg kg$^{-1}$) at least one week before influenza virus infection.

### Ganglion injection

NJP ganglia were injected with AAVs or diphtheria toxin as previously described[34], with minor modifications. Mice were anaesthetized (200 mg kg$^{-1}$ avertin, intraperitoneal injection), and depth of anaesthesia ensured throughout the procedure by lack of a toe pinch response. NJP ganglia were exposed and serially injected ($10 \times 13.8$ nl) with saline solution containing either AAVs (titre $> 6.7 \times 10^{12}$ vg ml$^{-1}$) or diphtheria toxin (5 µg ml$^{-1}$, Sigma) supplemented with 0.05% Fast Green FCF Dye (Sigma) using a sharply pulled glass pipette attached to a Nanoject Injector (Drummond). AAV-cre (AAV-Syn-Cre-GFP, SignaGen Laboratories, SL100892, AAV9), AAV-flex-tdTomato (AAV9.CAG.Flex.tdTomato. WPRE.BGH, Addgene, 51503, AAV9), AAV-GFP (pENN.AAV.CB7.CI.eGFP. WPRE.rBG, Addgene, 105542, AAV9), and AAV-flex-AP (AAV9.CAG.flex. PLAP.WPRE.bgH, custom virus, Boston Children's Hospital Viral Core, Boston, MA, AAV9) were purchased. Successful injection was verified by Fast Green FCF Dye slowly filling the entire ganglion without leakage. Surgical wounds were closed with coated VICRYL Sutures (J392H, Ethicon) and animals received Buprenorphine SR (BupSR-LAB, ZooPharm, 20 mg kg$^{-1}$ subcutaneously at the back of the neck) as an analgesic. Animals recovered for at least two weeks for behavioural analysis or four weeks for histological analysis.

### Nerve transection

Mice were anaesthetized (200 mg kg$^{-1}$ avertin, intraperitoneal injection), and depth of anaesthesia ensured throughout the procedure by lack of a toe pinch response. NJP ganglia were surgically exposed, and the glossopharyngeal nerve was identified and severed immediately adjacent to the ganglia using microscissors. Sham surgeries involved NJP ganglia exposure without nerve transection. Surgical wounds were closed with coated VICRYL Sutures (J392H, Ethicon) and animals received 20 mg kg$^{-1}$ Buprenorphine SR (BupSR-LAB, ZooPharm, subcutaneously at the back of the neck) as an analgesic. Animals recovered for at least two weeks before behavioural analysis.

### Histological and anatomical analysis

Immunohistochemistry and analysis of native tissue fluorescence was performed after intracardial perfusion of fixative (10 ml PBS, then 10 ml 4% paraformaldehyde in PBS). For whole-mount DTR immunostaining, NJP ganglia were collected and permeabilized (11.5 g glycine, 400 ml PBS with 0.2% Triton X-100, 100 ml DMSO) at 37 °C for 1 week. Ganglia were then blocked (1 h, room temperature) in blocking buffer (5% normal donkey serum, 017-000-121, Jackson in 0.05% Tween-20/PBS) and incubated (overnight, 4 °C) with 1:200 anti-DTR (HB-EGF, Goat, AF259NA, Fisher Scientific) in blocking buffer. Samples were washed ($4 \times 10$ min, 0.05% Tween-20 in PBS, room temperature) and incubated (2 h, room temperature) with anti-goat Alexa 488 solution (Jackson Immunoresearch, 705-545-147, 1:500 in PBS with 0.05% Tween-20). Tissues were washed again ($4 \times 10$ min, 0.05% Tween-20 in PBS, room temperature),

mounted (Fluoromount G medium, SouthernBiotech) onto microscope slides (Fisher Scientific), and visualized using a Leica SP5 II confocal microscope and analysed using ImageJ (1.53q). For cyclooxygenase-2 immunostaining, nasopharyngeal tissue was cryosectioned (15 µm), blocked (1 h, room temperature) with blocking solution (5% normal donkey serum, PBS with 0.01% Triton X-100), and incubated (overnight, 4 °C) with anti-COX2 antibody (rabbit, ab179800, Abcam, 1:500 in PBS with 0.01% Triton X-100). Sections were washed (3 × 10 min, room temperature, PBS with 0.01% Triton X-100) and incubated (2 h, room temperature) with anti-rabbit Alexa 488 solution (Jackson Immunoresearch, 711-545-152, 1:1,000 in PBS with 0.01% Triton X-100). Sections were washed again (3 × 10 min, room temperature, PBS with 0.01% Triton X-100), mounted with Fluoromount G medium, and visualized using a Leica SP5 II confocal microscope. To visualize brainstem axons, brainstem cryosections (20 µm) were obtained from AAV-injected mice, and processed as described above for cyclooxygenase-2 immunostaining in nasopharyngeal tissue, except antibodies were 1:1,000 anti-GFP (chicken, GFP-1020, Aves Labs), 1:1,000 anti-RFP (rabbit, 600-401-379, Rockland), anti-chicken Alexa 647 (Jackson Immunoresearch, 703-605-155, 1:300) and anti-rabbit Cy3 (Jackson Immunoresearch, 111-165-144, 1:300). To visualize axons in nasopharynx, fixed nasopharyngeal (15 µm) sections were stained as above for cyclooxygenase-2 immunostaining, but with anti-RFP primary antibody followed by anti-rabbit Cy3 secondary antibody. For GABRA1 neuron innervation to the trachealis and inferior pharyngeal constrictor muscles, tissues were collected fresh, cut along the ventral axis for open-book visualization, fixed (4% paraformaldehyde/PBS either 1 h, room temperature or overnight, 4 °C), and stained for tdTomato as above for tdTomato visualization in nasopharynx, except primary antibody incubation (36 h, 4 °C) and subsequent washes (3 × 12 h, 4 °C) were longer. For visualizing alkaline phosphatase, animals were perfused with PBS, and tissues were collected, fixed (4% paraformaldehyde, 1 h, room temperature), and washed in cold PBS. Tissues were then incubated (70 °C, 2 h) in alkaline phosphatase buffer (0.1 M Tris HCl pH 9.5, 0.1 M NaCl, 50 mM MgCl$_2$, 0.1% Tween-20, 5 mM levamisole) and washed twice in alkaline phosphatase buffer. Alkaline phosphatase activity was visualized with NCT/BCIP solution (34042, ThermoFisher Scientific) according to manufacturer's protocols. Stained samples were post-fixed (4% paraformaldehyde, overnight, 4 °C), dehydrated through a series of ethanol washes, and cleared using a 1:2 mixture of benzyl alcohol (Sigma-Aldrich 402834-500ML): benzyl benzoate (Sigma-Aldrich B6630-1L). The tissue was then cut along the ventral axis for open-book visualization. Whole-mount stainings were captured by microscopy with an AxioZoom (Zeiss) and analysed using ImageJ (1.53q).

### Analysis of mRNA expression in situ

DNA-based hybridization chain reaction (HCR) probes against *Ptger3* (lot no. PRH698) and *Phox2b* (lot no. PRF341) were purchased from Molecular Instruments. Hybridization solution (30% formamide, 5× SSC, 9 mM citric acid (pH 6), 0.1% Tween-20, 50 µg ml$^{-1}$ heparin, 1× Denhardt's solution, 10% dextran sulfate), probe wash buffer (30% formamide, 5× SSC, 9 mM citric acid pH 6.0, 0.1% Tween-20, 50 µg ml$^{-1}$ heparin), amplification buffer (5× SSC, 0.1% Tween-20, 10% dextran sulfate), and fluorophore-labelled HCR amplifiers were purchased from Molecular Instruments. NJP ganglia were freshly collected, mounted in OCT (Sakura Finetek) and cryosectioned (10 µm). Tissue was post-fixed (4% PFA, PBS, room temperature, 20 min), washed (3 × 10 min, PBS, room temperature), treated with 1% hydrogen peroxide (PBS, room temperature, 20 min), and incubated with *Ptger3* and *Phox2b* HCR probes (0.4 pmol, hybridization buffer in a humidified chamber, 37 °C, overnight). Tissue was then washed (15 min, 37 °C) with (1) 3:1 probe wash buffer: 5× SSCT (5× SSC, 0.1% Tween-20), (2) 1:1 probe wash buffer: 5× SSCT, (3) 1:3 probe wash buffer: 5× SSCT, and (4) 5× SSCT. Tissue was then incubated with amplification buffer (30 min, room temperature), and then with fluorescent HCR amplifiers (6 pmol, Alexa594 for *Ptger3* probe and Alexa 488

for *Phox2b* probe) in amplification buffer (humidified chamber, room temperature, overnight). Tissue was then washed (5× SSCT, 2 × 30 min, 1 × 5 min, room temperature), mounted in Fluoromount G medium, and visualized with by confocal microscopy (Leica SP5 II).

### Single-cell transcriptomics

All UMAP plots in this manuscript were made from published single-cell transcriptome data (GEO Accession ID: GSE145216)[22] using Seurat (4.1.0) and R Studio (4.1.2).

### Quantitative PCR analysis

Levels of mouse *Ptger3*, *Gapdh* transcript and the influenza virus nucleoprotein (NP) were measured in cDNA obtained from hypothalamus, NJP ganglia, nasal lavage (for upper respiratory tract analysis), and/or BALF (for lower respiratory tract analysis). BALF was collected by surgically opening the neck and cannulating a 21G catheter into the trachea. In total, 3 × 1 ml of PBS was slowly injected into the lungs then solution was gently aspirated back to the syringe and collected. Collected BALF was centrifuged (7 min, 400$g$, 4 °C), and the supernatants were stored (−80 °C) until analysis. Nasal lavage was performed by inserting a needle (26G) into the upper trachea angled toward the nose. Sterile PBS (1 ml) was administered through the needle via a cannulated syringe (PE20 polyethylene cannula, inner diameter 0.38 mm), and collected after expulsion from the nose. RNA was collected and cDNA synthesized from BALF, nasal lavage fluid, and homogenized tissue using Trizol (15596026, ThermoFisher) and the High-Capacity cDNA Reverse Transcription Kit (4368813, ThermoFisher) according to the manufacturer's protocols. qPCR analysis was performed on cDNA using gene-specific primers, PowerTrack SYBR Green Master Mix (A46012, ThermoFisher) on QuantStudio 7 qPCR instrument (ThermoFisher). *Gapdh* primers were GGGTGTGAACCACGAGAAATATG and TGTGAGGGAGATGCTCAGTG TTG; *Ptger3* primers were CAACCTTTTCTTCGCCTCTG and TTTCTG CTTCTCCGTGTGTG; and influenza virus nucleoprotein primers were GACGATGCAACGGCTGGTCTG and ACCATTGTTCCAACTCCTTT. Expression levels of *Ptger3* and nucleoprotein were normalized to *Gapdh* levels. Normalization was by the $2^{-\Delta CT}$ method for Extended Data Fig. 4a, and the $2^{-\Delta\Delta CT}$ method[53] for Extended Data Fig. 9b, with added normalization using values from uninfected controls.

### PGE2 and cytokine measurements

Mice were euthanized at different time points after infection with influenza A virus. Whole blood was collected by cardiac puncture using a 0.5 M EDTA-coated needle and syringe and mixed immediately with 0.5 M EDTA (50 µl). Plasma was isolated by centrifugation (3,000 rpm, 10 min, 4 °C) and the supernatant stored (−80 °C) until ELISA analysis. BALF was collected as described above. PGE2 and cytokine levels were measured in plasma and/or BALF by ELISA (R&D Systems, PGE2: KGE004B, TNF: DY410-5, IFNγ: DY485-05, IL-6: DY406-05) according to the manufacturer's instructions.

### Calcium imaging

NJP ganglia were acutely collected from *Gabra1-IRES-cre; lsl-tdTomato* mice and incubated (37 °C, 90 min, with rotation) in dissociation solution (Dulbecco's Modified Eagle's Medium (DMEM) containing liberase (55 mg ml$^{-1}$, Roche, 05401135001), DNase (0.004%, Worthington, LS002007)). Cells were than pelleted (3 min, 300$g$, 4 °C) and resuspended in 0.2% ovomucoid, 0.004% DNase, DMEM, 200 µl. Using a P200 pipette, cells were triturated at least 10 times until clumps were not visible, filtered through a 40 µm mesh cell strainer, and pelleted again (3 min, 300$g$, 4 °C). Cells were then resuspended in culture medium containing 5 µM Calbryte 520 AM (20651, AAT Bioquest); culture medium contains 1× B27 (Gibco, 17504044), 1× N2 (Gibco, 17502048), 1× penicillin and streptomycin (Gibco, 15140122) in Neurobasal media without phenol red (Gibco, 12348017). Resuspended cells were then plated onto laminin-coated cover glass and incubated (37 °C, at least 30 min

before imaging) in a humidified $CO_2$ incubator. The cover glass was then transferred to a chamber with an inlet and outlet perfusion (Warner Instruments, RC-24N) with continuous flow of Hank's balanced salt solution (HBSS). The calcium responses were then imaged (488 nm) on perfusion of prostaglandin E2 (1 µM in HBSS, 2 min) and KCl (150 mM, at the end of the run). GABRA1-positive neurons were identified by tdTomato fluorescence (568 nm). Cells were identified as responsive if mean Calbryte 520 AM fluorescence during the stimulation period exceeded three standard deviations above the baseline mean.

## Fibre photometry

Fibre photometry of AGRP neurons was performed as described[54] with PGE2 (intraperitoneal injection, 0.5 mg kg$^{-1}$) or PBS acutely injected during a brief manual restraint using Synapses software (Build: 94-42329P, Tucker-Davis Technology). Data were analysed using Matlab (R2020a).

## Statistical analysis

Data in graphs are represented as mean ± s.e.m., with all sample sizes and $P$ values provided in figure legends. Statistical analysis for behavioural experiments involved averaging daily changes in parameters indicated per mouse (days 1–10 after infection or through survival). Statistical significance was calculated with Prism 8.4.3 software (GraphPad), and involved statistical tests described in figure legends. Probit analysis in Extended Data Fig. 5b was performed with SPSS, 21 (IBM). Sample sizes were chosen based on prior expertise and publications in our field[26,55] and are disclosed in the Figure legends. Animal groups were randomly assigned and control animals were age-matched to experimental animals. The same investigator performed genotyping and analysis of sickness responses, so data were not generated blind to genotype or experimental group.

## Reporting summary

Further information on research design is available in the Nature Portfolio Reporting Summary linked to this article.

## Data availability

All data used to generate figures, including behavioural data points from each individual mouse are provided as source data. All reagents that may not be commercially available including certain transgenic mice and AAVs will be made freely available upon reasonable request. Source data are provided with this paper.

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

**Acknowledgements** We thank C. Saper for *Ptger3$^{flox}$* mice and comments on the manuscript. Cartoons were created using BioRender.com. This work was supported by grants from the NIH (DP1 AT009497 and R01 HL132255) and the Chan Zuckerberg Initiative to S.D.L., a Banting Postdoctoral Fellowship to N.-R.B. and a Harvard Medical School Goldberg Fellowship to N.H. S.L.P. was a Warren Alperts Distinguished Scholar and S.D.L. is an investigator of the Howard Hughes Medical Institute.

**Author contributions** N.-R.B., S.L.P., N.H., I.M.C. and S.D.L. designed experiments. N.-R.B., N.H., S.L.P. and Y.W. performed experiments. N.-R.B., S.L.P., N.H., I.M.C. and S.D.L. analysed data. N.-R.B. and S.D.L. wrote the manuscript.

**Competing interests** S.D.L. is a consultant for Kallyope, Inc. The other authors declare no competing interests.

**Additional information**
**Correspondence and requests for materials** should be addressed to Stephen D. Liberles.

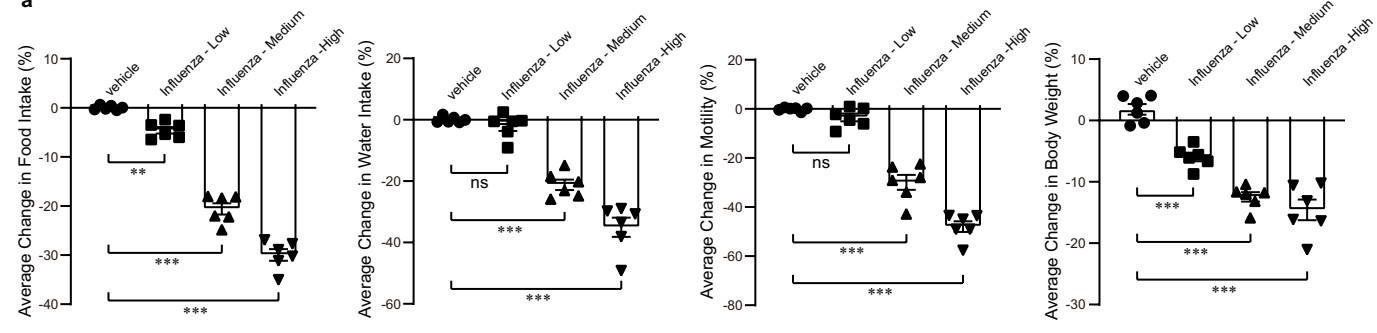

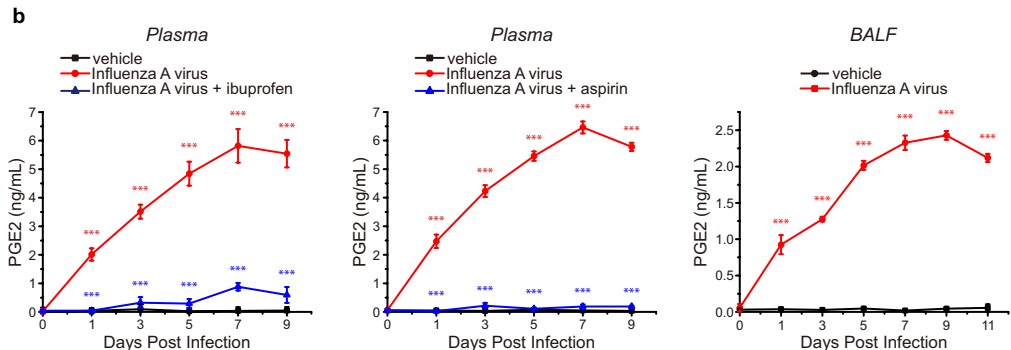

**Extended Data Fig. 1 | Influenza A infection induces behavioral changes and PGE2 production. a**, Statistical analysis for behavioral experiments involved averaging daily changes in parameters indicated per mouse (days 1-10 after infection or through survival), as represented here in bar graphs. Statistical values are identical to those in Fig. 1a. **b**, PGE2 levels in plasma (left, middle) and BALF (right) were measured by ELISA at time points indicated after exposure to influenza A virus (red, blue) or vehicle control (black). Some

virus-infected mice (blue) were additionally given *ad libitum* access to ibuprofen (1 mg/ml) in drinking water from 3 days prior to infection (left) or received daily aspirin administration (IP, 20 mg/kg), mean ± sem, n: 3 mice per group, ***p < 0.0005 by two-way ANOVA followed by Bonferroni's multiple comparison test with comparisons made between red and black curves (red stars) or red and blue curves (blue stars). p values in **b** are <0.0001 for all indicated stars.

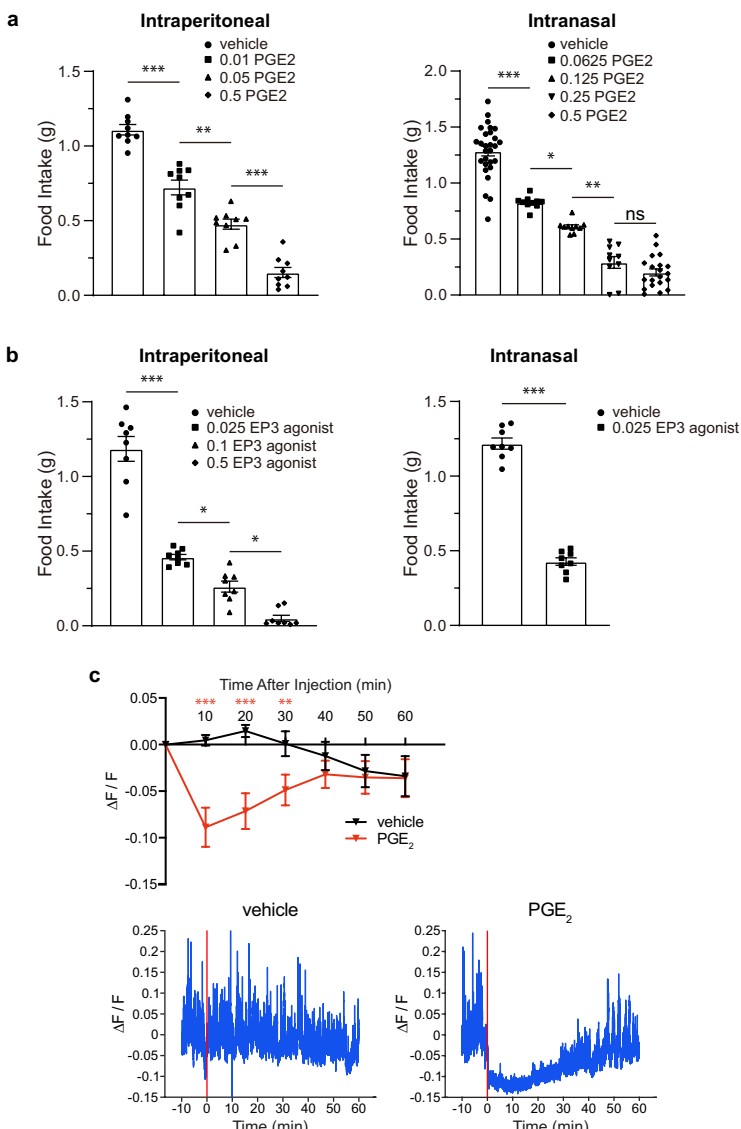

**Extended Data Fig. 2 | PGE2 decreases feeding and AGRP neuron activity.**
**a**, Food intake by fasted mice administered with PGE2 (left: IP, right: intranasal) at doses indicated (mg/kg), and given ad libitum access to food (1 h), mean ± sem, n: 9 (left) mice per group, 28 (vehicle), 10 (0.0625, 0.125, 0.25), 20 (0.5) in (right), ***p < 0.0005, **p < 0.005, *p < 0.05, ns: not significant by two-way ANOVA Tukey's multiple comparison test. **b**, Food intake by fasted mice administered with the EP3 receptor agonist sulprostone (left: IP, right: intranasal) at doses indicated (mg/kg), and given ad libitum access to food (1 h), mean ± sem, n: 8 mice per group, ***p < 0.0005, *p < 0.05 by two-way ANOVA Tukey's multiple comparison test (left) or two-tailed unpaired t-test (right).

**c**, GCaMP6s fluorescence (ΔF/F) was measured in AGRP neurons of the arcuate nucleus by fiber photometry before and after IP injection of PGE2 (0.5 mg/kg in PBS) or vehicle alone (PBS). (top) Responses are depicted as the mean of measurements made in 10-minute time intervals (for example, 10 refers to the mean of measurements made between 0 and 10 min), mean ± sem, n: 12 mice per group, **p < 0.01, ***p < 0.001 by two-way ANOVA with Bonferroni's multiple comparison test, (bottom) representative recording traces with red bar indicating time of injection. p values left to right in **a**: IP: <0.0001, 0.0005, <0.0001, intranasal: <0.0001, 0.0455, 0.0008, 0.6630; **b**, IP: <0.0001, 0.0312, 0.0187, intranasal: <0.0001; **c**: <0.0001, <0.0001, 0.0045.

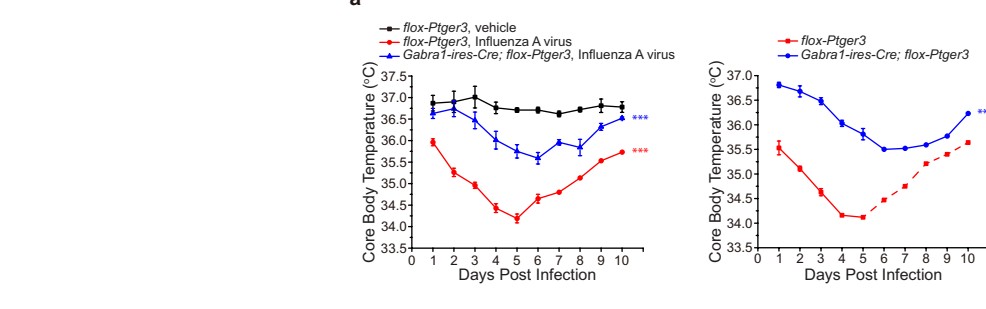

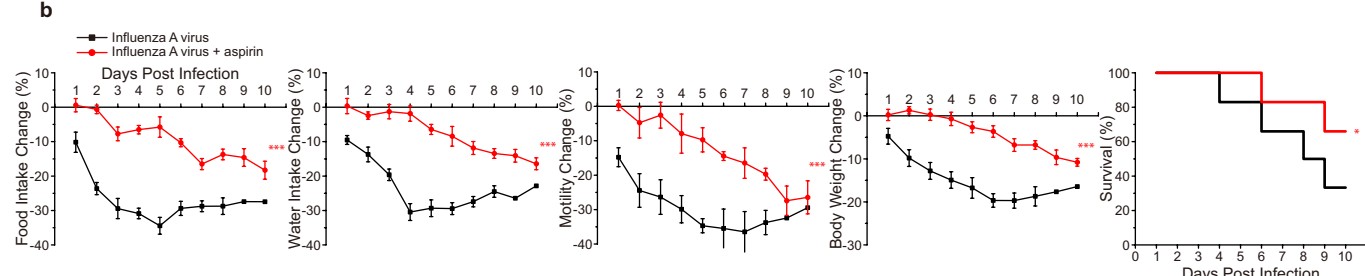

**Extended Data Fig. 3 | Influenza-induced hypothermia and sickness behaviors are attenuated by cell-specific *Ptger3* knockout and aspirin.**
**a**, Core body temperature in *flox-Ptger3* and *Gabra1-ires-Cre; flox-Ptger3* mice was measured daily after administration of influenza virus (red, blue) or saline (black) using a rectal probe (left) or radiotelemetry (right), mean ± sem, n: 6 (left) and 3 (right) mice per group, ***p < 0.0005 by one-way ANOVA Dunnett's multiple comparison test (left); ***p < 0.0005 by two-tailed unpaired t-test for right as detailed in Fig. 1 for behavior/physiology analysis (right), comparisons between red and black curves (red stars) or between red and blue curves (blue stars). For data on the right, two of three control *flox-Ptger3* mice died on day 5,

so subsequent data is represented as a dashed line and statistical analysis was performed only on data from days 1-4. **b**, Mice were given daily injections of aspirin (IP, 20 mg/kg), red or vehicle alone (black) throughout the paradigm. After three days, mice were infected with influenza A virus and subsequently monitored as indicated, mean ± sem, n: 6 mice per group, ***p < 0.0005 by two-tailed unpaired t-test as detailed in Fig. 1 for behavior/physiology analysis, *p < 0.05 by a log-rank (Mantel-Cox) test for survival analysis. p values in **a**, left: red <0.0001, blue 0.0003, right: <0.0001; **b**, left to right: <0.0001, <0.0001, <0.0001, <0.0001, 0.0243.

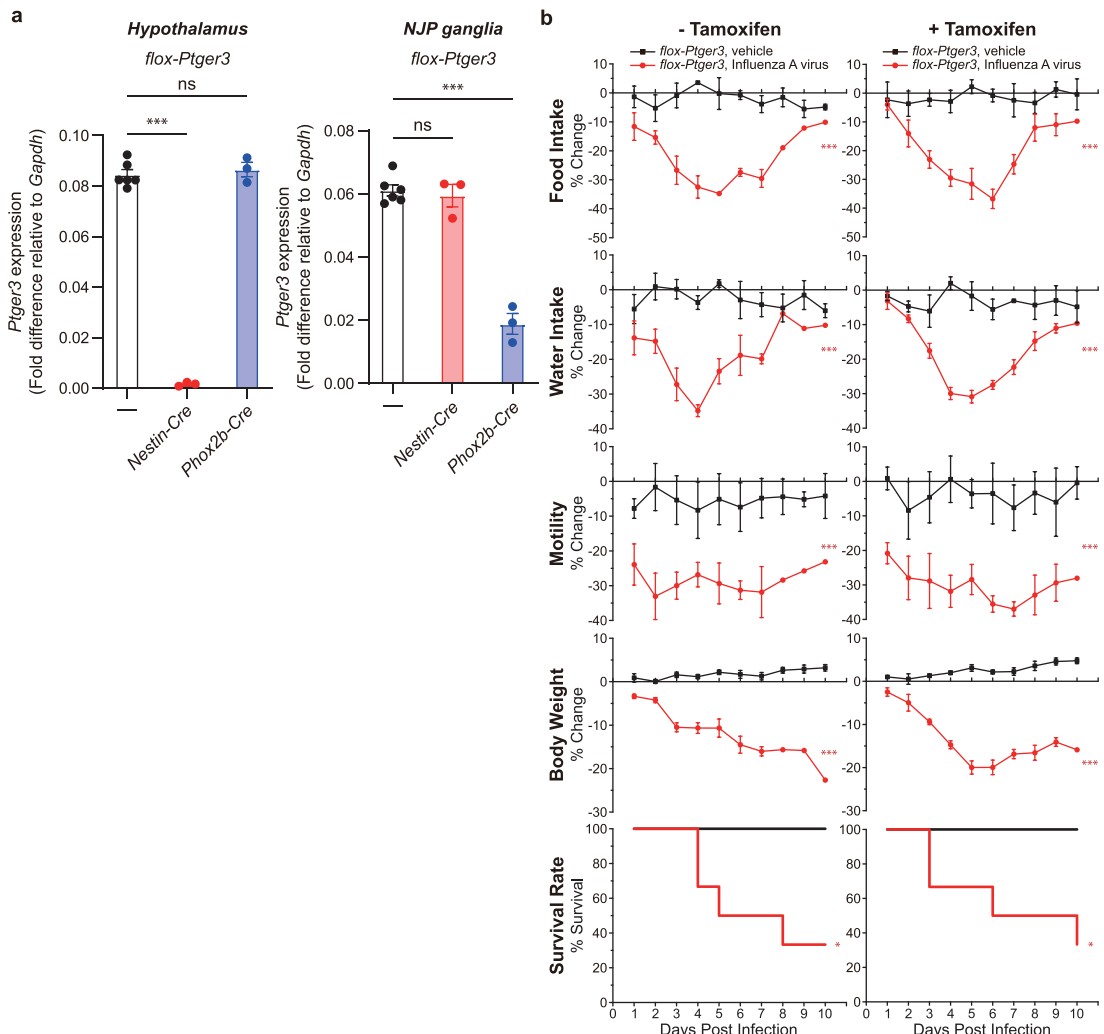

**Extended Data Fig. 4 | Validating approaches for cell-specific *Ptger3* knockout. a**, qPCR analysis of *Ptger3* expression in hypothalamus (left) or NJP ganglia (right) of *flox-Ptger3* (white), *Nestin-Cre;flox-Ptger3* (red), and *Phox2b-Cre;flox-Ptger3* (blue) mice. *Phox2b-Cre* mice display Cre expression in nodose and petrosal but not jugular neurons[56]. *Ptger3* transcript levels were expressed after normalization to *Gapdh* expression levels, mean ± sem, n: 6 mice for *flox-Ptger3* and 3 for other groups, ***p < 0.0005, ns: not significant by one-way ANOVA Dunnett's multiple comparison test. **b**, *flox-Ptger3* control mice were treated with (right) or without (left) tamoxifen for 5 consecutive days (IP, 70 mg/kg) as was done for *Advillin-Cre^ER;flox-Ptger3* mice. At least one week later, mice were either exposed to influenza A virus (red) or saline (black) and monitored as indicated, mean ± sem, n: 6 mice per group, ***p < 0.0005 by two-tailed unpaired t-test as detailed in Fig. 1 for behavior/physiology analysis, *p < 0.05 by a log-rank (Mantel-Cox) test for survival analysis. p values in **a**, left: <0.0001, 0.7506, right: 0.8680, <0.0001; **b** top to bottom, left: <0.0001, <0.0001, <0.0001, <0.0001, 0.0179, right: <0.0001, 0.0003, <0.0001, 0.0179.

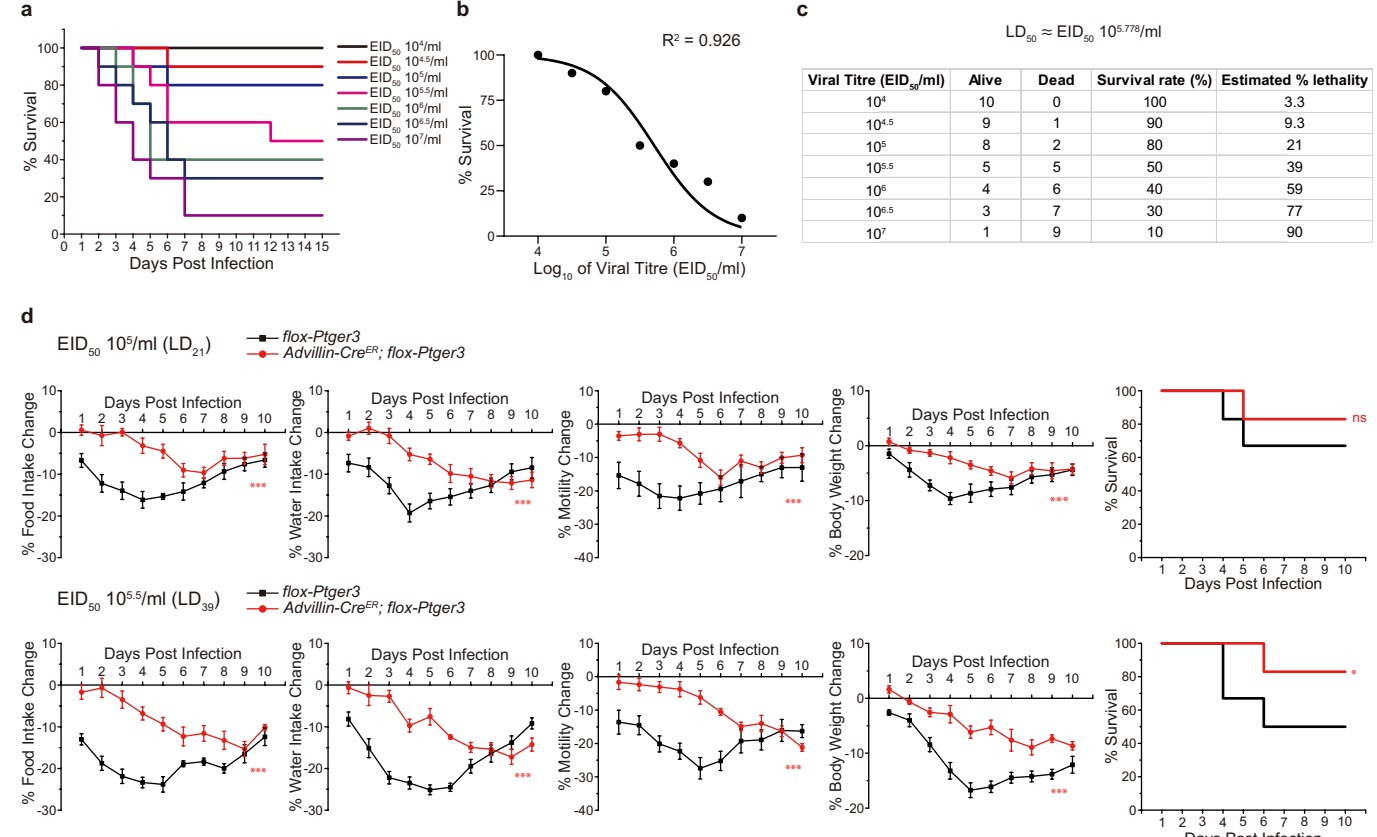

**Extended Data Fig. 5 | *Ptger3* knockout in peripheral sensory neurons attenuates behavioral responses to sublethal influenza A doses. a**, Wild type mice were infected with influenza A virus at titers indicated and subsequent survival monitored daily, n: 10 per group. **b**, a dose-response curve of viral titer (log$_{10}$) vs. survival rates with non-linear fit of $R^2 = 0.926$. **c**, A table indicating survival rates to various influenza A virus inoculation doses, with the estimated % lethality determined by Probit regression analysis (Statistical Product and Service Solutions). **d**, *Advillin-Cre^{ER}; flox-Ptger3* mice (red,

previously injected with tamoxifen) or *flox-Ptger3* mice (black) indicated were infected with sublethal doses of influenza A virus (top: $10^5$ EID$_{50}$ or LD$_{21}$, bottom: $10^{5.5}$ EID$_{50}$ or LD$_{39}$) and monitored daily as indicated, mean ± sem, n: 6 mice per group, ***p < 0.0005 by two-tailed unpaired t-test as detailed in Fig. 1 for behavior/physiology analysis, *p < 0.05, ns: not significant by a log-rank (Mantel-Cox) test for survival analysis. p values left to right in **d**, top: <0.0001, <0.0001, 0.0002, <0.0001, 0.4788; bottom: <0.0001, <0.0001, 0.0004, 0.0004, 0.0078.

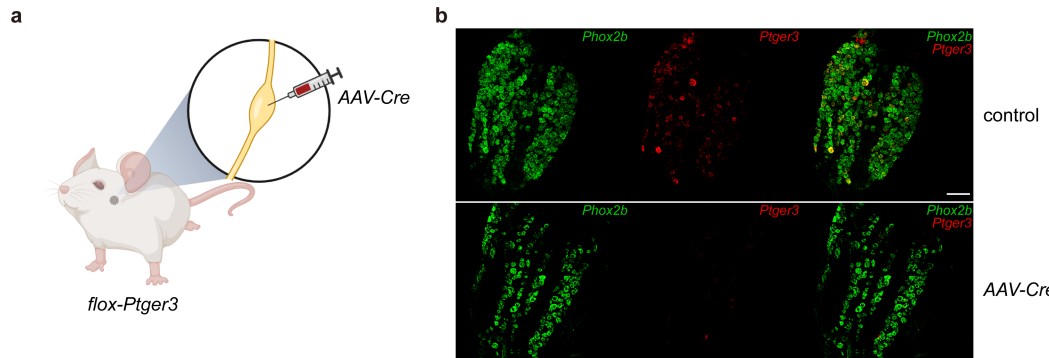

**Extended Data Fig. 6 | Validating AAV-driven *Ptger3* knockout in NJP ganglia. a**, Cartoon depicting bilateral injection of *AAV-Cre* into NJP ganglia of *flox-Ptger3* mice. **b**, The NJP ganglia of *flox-Ptger3* mice were injected bilaterally with *AAV-Cre* (bottom) or saline (control, top), and cryosections of NJP ganglia were subsequently examined by two-color RNA *in situ* hybridization to detect *Phox2b* (green) and *Ptger3* (red), scale bar: 100 μm. Images are representative from three independent experiments. Part **a** created with BioRender.com.

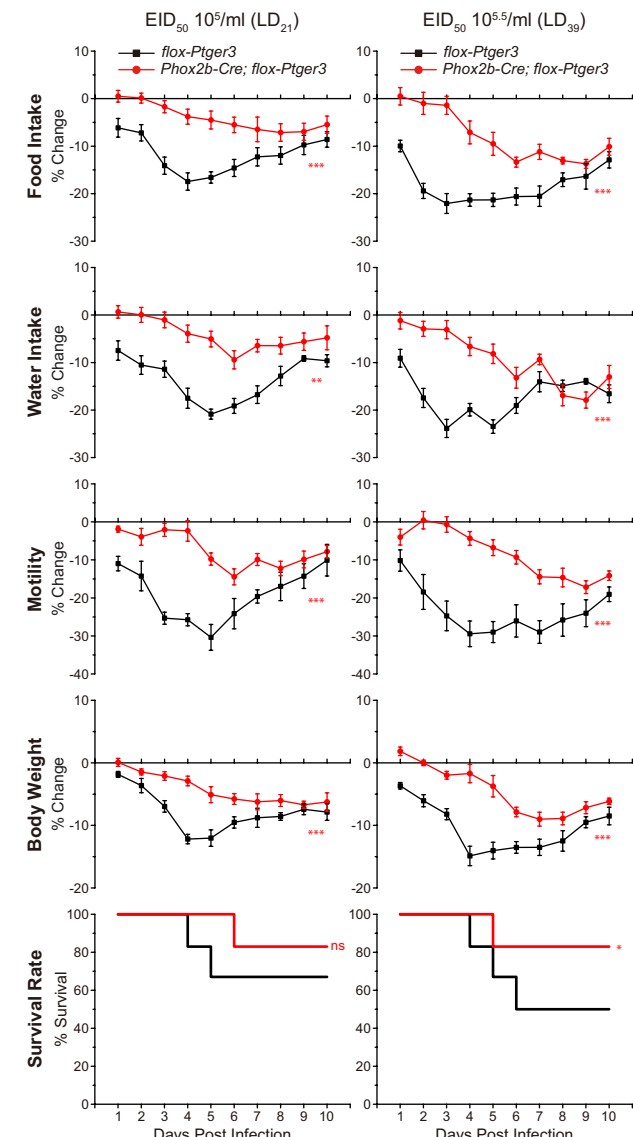

**Extended Data Fig. 7 | Cell-specific *Ptger3* knockout attenuates behavioral responses to sublethal influenza A doses.** *Phox2b-Cre;flox-Ptger3* mice (red) or *flox-Ptger3* mice (black) were infected with sublethal doses of influenza A virus (top: $10^5$ EID$_{50}$ or LD$_{21}$, bottom: $10^{5.5}$ EID$_{50}$ or LD$_{39}$) and monitored daily as indicated, mean ± sem, n: 6 mice per group, **p < 0.005, ***p < 0.0005 by two-tailed unpaired t-test as detailed in Fig. 1 for behavior/physiology analysis, *p < 0.05, ns: not significant by a log-rank (Mantel-Cox) test for survival analysis. p values top to bottom left: <0.0001, <0.0001, <0.0001, 0.0002, 0.4524; right: <0.0001, <0.0001, <0.0001, 0.0002, 0.0155.

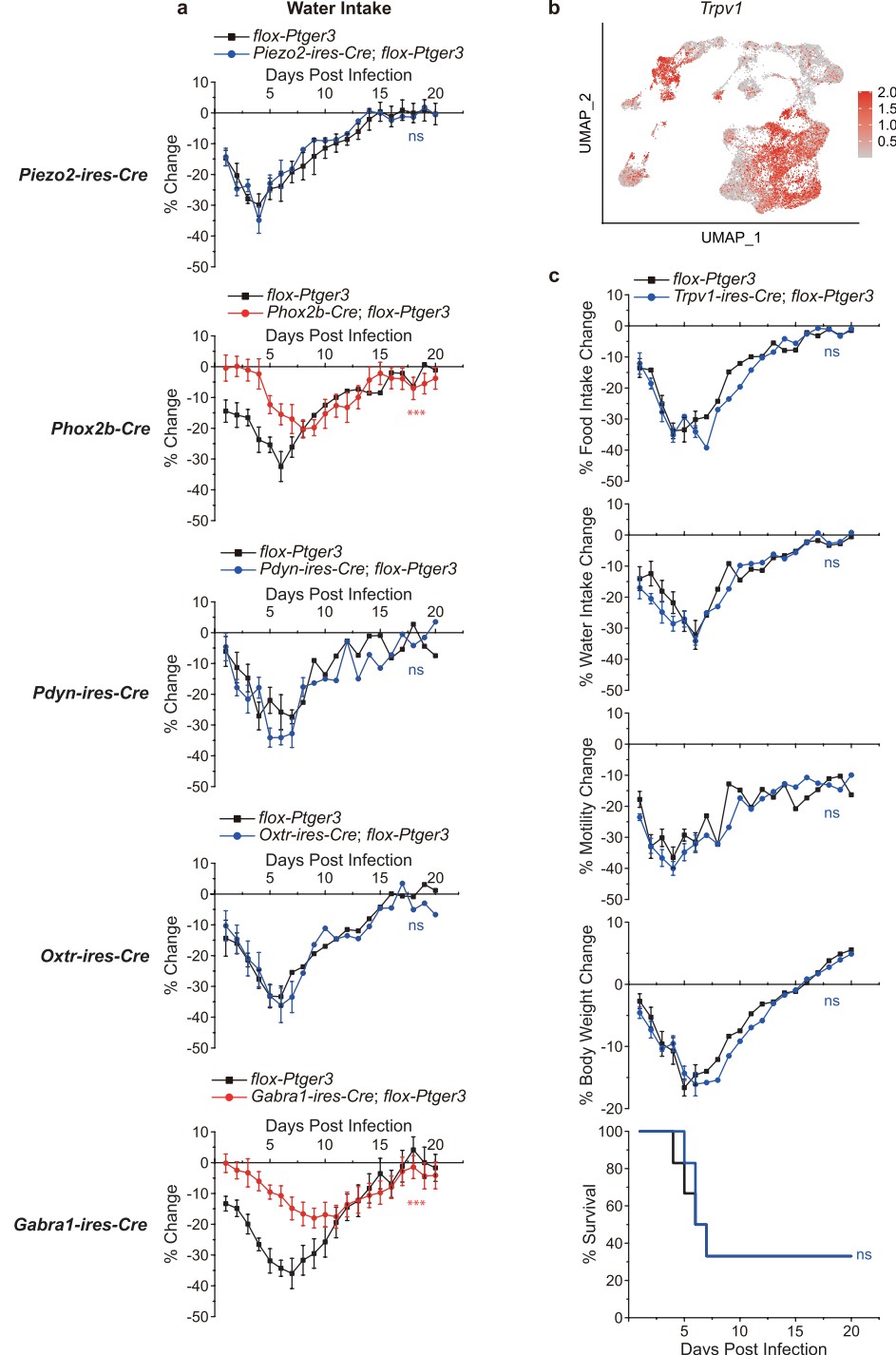

**Extended Data Fig. 8 | Measures of flu-induced behavioral changes in cell-specific *Ptger3* knockouts. a**, Water intake changes after flu infection in mice of Fig. 3, mean ± sem, n: 8 (*Piezo2-ires-Cre*), 6-8 (*Phox2b-Cre*; 8 control and 6 *Phox2b-Cre*), 6 (*Pdyn-ires-Cre*), 6 (*Oxtr-ires-Cre*), and 10 (*Gabra1-ires-Cre*) mice per group, ***p < 0.0005, ns: not significant by two-tailed unpaired t-test as detailed in Fig. 1 for behavior/physiology analysis. **b**, A Uniform Manifold Approximation and Projection (UMAP) plot derived from published single-cell transcriptome data of vagal and glossopharyngeal sensory ganglia[22] indicating

*Trpv1* expression (red shading: natural log scale). **c**, *Trpv1-ires-Cre; flox-Ptger3* mice (blue) or *flox-Ptger3* mice (black) were infected with influenza A virus and monitored daily as indicated, mean ± sem, n: 6 mice per group, ns: not significant by two-tailed unpaired t-test as detailed in Fig. 1 for behavior/physiology analysis, ns: not significant by a log-rank (Mantel-Cox) test for survival analysis. p values top to bottom in **a**: 0.9101, 0.0004, 0.0534, 0.9544, <0.0001 and **c**: 0.2485, 0.0705, 0.0888, 0.1801, 0.8412.

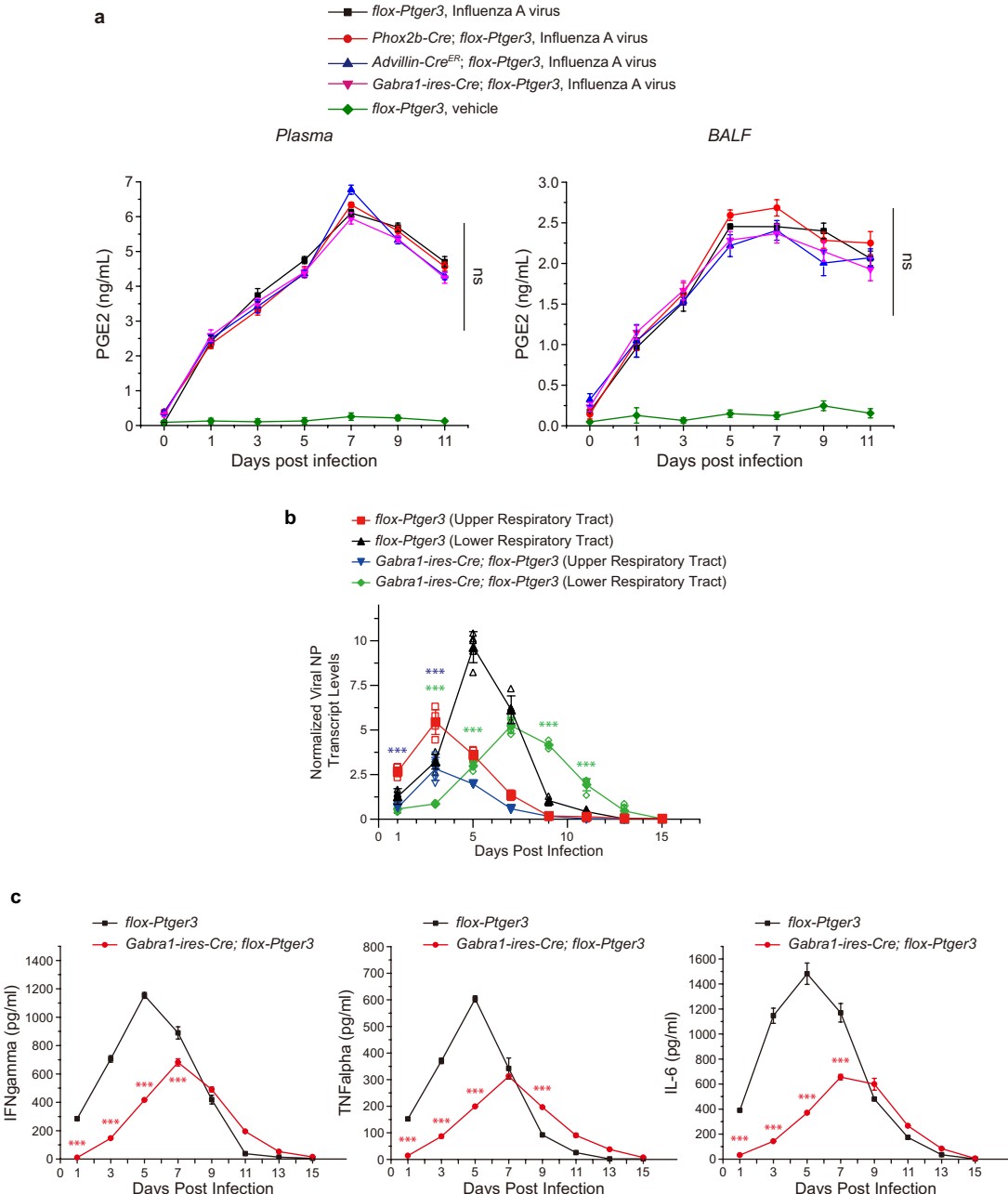

**Extended Data Fig. 9 | Impact of targeted *Ptger3* knockout on cytokines and viral transcript levels. a**, PGE2 levels in plasma (left) and BALF (right) were measured by ELISA in mice indicated at various time points after exposure to influenza A virus or PBS control, mean ± sem, n: 5 mice per group, ns: not significant by two-way ANOVA involving analysis of influenza virus-infected groups. **b**, qPCR analysis of viral *nucleoprotein* (NP) transcript levels in the upper and lower respiratory tract of *flox-Ptger3* and *Gabra1-ires-Cre; flox-Ptger3* mice after influenza virus infection, normalized to *Gapdh* and uninfected controls, mean ± sem, n: 5 mice per group, ***p < 0.0005 by two-way ANOVA followed by Bonferroni's multiple comparison test with comparisons made between red and blue curves (blue stars) or black and green curves (green stars). **c**, Levels of IFNγ, TNFα, and IL-6 in BALF were measured by ELISA at time points indicated after exposure to influenza A virus in *flox-Ptger3* (black) or *Gabra1-ires-Cre; flox-Ptger3* (red), mean ± sem, n: 3 mice per group, ***p < 0.0005 by two-way ANOVA followed by Bonferroni's multiple comparison test with comparisons made between red and black curves (red stars). p values in **a**: 0.1591, 0.0662, **b** and **c**: <0.0001 for all indicated stars.

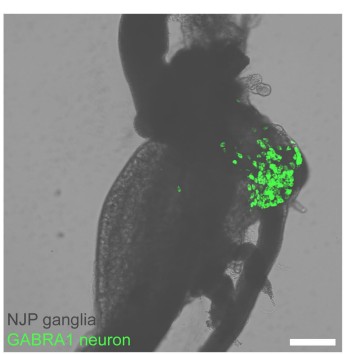

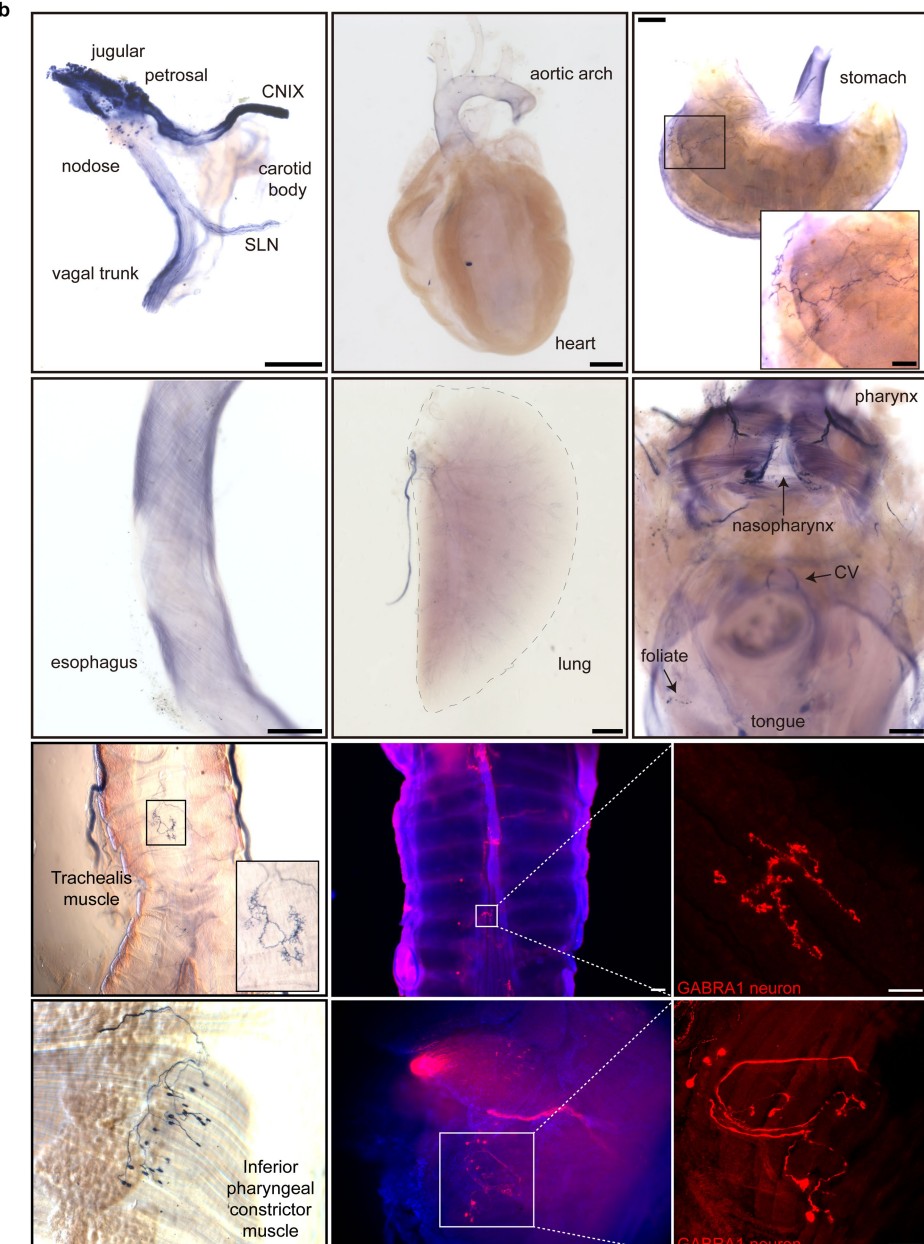

**Extended Data Fig. 10 | Sparse innervation of internal organs by GABRA1 NJP neurons. a**, Native GFP fluorescent signals in wholemount preparations of NJP ganglia from *Gabra1-ires-cre; lsl-L10 GFP*, scale bar: 200 μm. **b**, NJP ganglia of *Gabra1-ires-Cre* mice were injected bilaterally with Cre-dependent *AAV-flex-AP* or *AAV-flex-tdTomato* and axons were visualized in fixed wholemount tissue preparations using either a colorimetric alkaline phosphatase substrate (top two rows, left images in bottom two rows) or tdTomato immunostaining (right two images in bottom two rows). Scale bars (left to right) top row: 500, 1000, 1000 (inset: 250) μm; 2nd row: 500, 1000, 500 μm; 3rd row: 200, 50 μm; bottom row: 200, 100 μm. Images are representative of three independent experiments involving GFP and tdTomato and two independent experiments involving alkaline phosphatase.

# Reporting Summary

## Statistics

For all statistical analyses, confirm that the following items are present in the figure legend, table legend, main text, or Methods section.

| n/a | Confirmed | |
|---|---|---|
| ☐ | ☒ | The exact sample size (*n*) for each experimental group/condition, given as a discrete number and unit of measurement |
| ☐ | ☒ | A statement on whether measurements were taken from distinct samples or whether the same sample was measured repeatedly |
| ☐ | ☒ | The statistical test(s) used AND whether they are one- or two-sided<br>*Only common tests should be described solely by name; describe more complex techniques in the Methods section.* |
| ☒ | ☐ | A description of all covariates tested |
| ☐ | ☒ | A description of any assumptions or corrections, such as tests of normality and adjustment for multiple comparisons |
| ☐ | ☒ | A full description of the statistical parameters including central tendency (e.g. means) or other basic estimates (e.g. regression coefficient) AND variation (e.g. standard deviation) or associated estimates of uncertainty (e.g. confidence intervals) |
| ☐ | ☒ | For null hypothesis testing, the test statistic (e.g. *F*, *t*, *r*) with confidence intervals, effect sizes, degrees of freedom and *P* value noted<br>*Give P values as exact values whenever suitable.* |
| ☒ | ☐ | For Bayesian analysis, information on the choice of priors and Markov chain Monte Carlo settings |
| ☒ | ☐ | For hierarchical and complex designs, identification of the appropriate level for tests and full reporting of outcomes |
| ☒ | ☐ | Estimates of effect sizes (e.g. Cohen's *d*, Pearson's *r*), indicating how they were calculated |

*Our web collection on statistics for biologists contains articles on many of the points above.*

## Software and code

Policy information about availability of computer code

| Data collection | Leica Confocal LAS AF for confocal images. Zeiss ZEN for alkaline phosphatase images. Logitech Capture (2.08.11) for motility recordings. Synapses software (Build: 94-42329P, Tucker-Davis Technology) for fiber photometry recordings. iox software (2.10.8, Emka Technologies) for telemetry based body temperature recordings. |
|---|---|
| Data analysis | ImageJ (1.53q), R Studio (4.1.2), Seurat (4.1.0), Matlab (R2020a), GraphPad Prism (8.4.3), and SPSS (21) were used for for data analysis. |

For manuscripts utilizing custom algorithms or software that are central to the research but not yet described in published literature, software must be made available to editors and reviewers. We strongly encourage code deposition in a community repository (e.g. GitHub). See the Nature Portfolio guidelines for submitting code & software for further information.

## Data

Policy information about availability of data

All manuscripts must include a data availability statement. This statement should provide the following information, where applicable:
- Accession codes, unique identifiers, or web links for publicly available datasets
- A description of any restrictions on data availability
- For clinical datasets or third party data, please ensure that the statement adheres to our policy

All data used to generate figures, including behavioral data points from each individual animal, are provided as source data. Gene expression analysis involved previously published and already fully available single cell RNA sequencing data, and the accession number for that data (GEO:GSE145216) is now reported in this paper.

# Field-specific reporting

Please select the one below that is the best fit for your research. If you are not sure, read the appropriate sections before making your selection.

☒ Life sciences          ☐ Behavioural & social sciences          ☐ Ecological, evolutionary & environmental sciences

For a reference copy of the document with all sections, see nature.com/documents/nr-reporting-summary-flat.pdf

# Life sciences study design

All studies must disclose on these points even when the disclosure is negative.

| | |
|---|---|
| Sample size | Sample sizes were chosen based on prior expertise and publications in our field (for example, Baral et al., Nature Medicine, 2018, Ilanges et al., Nature, 2022) and are disclosed in each figure legend. |
| Data exclusions | No data were excluded from the analysis. |
| Replication | All experiments where representative images were depicted were independently replicated at least twice, and typically three times, as detailed for each experiment in figure legends. |
| Randomization | Animals were randomly assigned to experimental cohorts, based only genotype and appropriate age-matching. |
| Blinding | The same investigator performed genotyping and analysis of sickness responses, so data were not generated blind to genotype or experimental group. Nonetheless, all animals in each experiment were analyzed without bias. |

# Reporting for specific materials, systems and methods

We require information from authors about some types of materials, experimental systems and methods used in many studies. Here, indicate whether each material, system or method listed is relevant to your study. If you are not sure if a list item applies to your research, read the appropriate section before selecting a response.

## Materials & experimental systems

| n/a | Involved in the study |
|---|---|
| ☐ | ☒ Antibodies |
| ☒ | ☐ Eukaryotic cell lines |
| ☒ | ☐ Palaeontology and archaeology |
| ☐ | ☒ Animals and other organisms |
| ☒ | ☐ Human research participants |
| ☒ | ☐ Clinical data |
| ☒ | ☐ Dual use research of concern |

## Methods

| n/a | Involved in the study |
|---|---|
| ☒ | ☐ ChIP-seq |
| ☒ | ☐ Flow cytometry |
| ☒ | ☐ MRI-based neuroimaging |

## Antibodies

| | |
|---|---|
| Antibodies used | Chicken anti-GFP primary antibody- Aves Labs, Cat number: GFP-1020, The Antibody Registry ID: AB_10000024.<br>Rabbit anti-RFP primary antibody- Rockland, Cat number: 600-401-379, The Antibody Registry ID: AB_2209751.<br>Goat anti-DTR primary antibody- R&D Systems, Cat number: AF-259-NA, The Antibody Registry ID: AB_354429.<br>Rabbit anti-COX2 primary antibody- Abcam, Cat number: ab179800, The Antibody Registry ID: AB_2894871.<br>Anti-goat Alexa488 secondary antibody- Jackson Immunoresearch, Cat number: 705-545-147.<br>Anti-rabbit Alexa488 secondary antibody- Jackson Immunoresearch, Cat number: 711-545-152.<br>Anti-rabbit Cy3 secondary antibody- Jackson Immunoresearch, Cat number: 111-165-144.<br>Anti-chicken Alexa647 secondary antibody- Jackson Immunoresearch, Cat number: 703-605-155. |
| Validation | Primary antibodies (rabbit anti-COX2 antibody, goat anti-DTR antibody, chicken anti-GFP antibody, rabbit anti-RFP antibody are commercially available, extensively used in prior studies,and for anti-COX2 validated in knockout animals by the manufacturer. In our previous work with anti-DTR, GFP, and RFP antibodies, background staining was not observed in wild type animals lacking antigen. |

## Animals and other organisms

Policy information about studies involving animals; ARRIVE guidelines recommended for reporting animal research

| | |
|---|---|
| Laboratory animals | Animals were maintained under constant temperature (23 ± 1°C) and relative humidity (46 ± 5%) with a 12-h light/dark cycle. Wild-type C57BL/6J (000664), Nestin-Cre (003771), Phox2b-Cre (016223), Advillin-CreER (032027), Trpv1-Cre (017769), Pdyn-ires-Cre (027958), Piezo2-ires-Cre (027719), Oxtr-ires-Cre (031303), Agrp-Cre (012899), lsl-Dtr (007900), lsl-tdTomato (Ai14, 007914) were |

purchased from Jackson Laboratories. Gabra1-ires-Cre, lsl-L10-Gfp, and flox-Ptger3 mice were previously generated (citations provided in text). There was no sex bias and the age of the animals was 6-8 weeks.

Wild animals                No wild animals were used.

Field-collected samples     No field-collected samples were used.

Ethics oversight            All animal procedures followed the ethical guidelines outlined in the NIH Guide for the Care and Use of Laboratory Animals, and all protocols were approved by the institutional animal care and use committee (IACUC) at Harvard Medical School.

Note that full information on the approval of the study protocol must also be provided in the manuscript.

