## [Peer Review File · Nature]

Manuscript Title: An airway-to-brain sensory pathway mediates influenza-induced sickness

Reviewer Comments & Author Rebuttals

Reviewer Reports on the Initial Version:

Referees' comments:

Referee #1 (Remarks to the Author):

Sickness behaviors is a stereotypic response to acute infections and its role in host defense has been a subject of fascination but relatively little understanding of the underlying mechanisms. Here, the authors used state of the art neuroscience approaches to identify neuronal pathways responsible for the sickness behaviors induced by influenza infection. The study dramatically clarifies the contribution of central and peripheral neurons and identifies a small population of glossopharyngeal sensory neurons that are responsible for sickness behaviors in flu infection. These neurons express EP3 receptor, which is essential for orchestrating the behavioral response and change in homeostatic set points during illness. The study is very rigorous and conclusions are well supported and definitive. One interesting implication of the study is that different EP3 expressing sensory neurons that monitor different organ systems may be responsible for induction of sickness behaviors. This is important because it suggests that perhaps not all sickness behaviors are the same, which would certainly make physiological sense. It is also interesting that blocking sickness behaviors actually promoted survival if flu infections. There are maybe multiple possible explanations to this, including aspects of SPF conditions of the vivarium, or other ecological or evolutionary reasons that would be interesting to explore in the future.

In conclusion, this study is really a milestone in understanding sickness behaviors. It will provide the impetus for further studies examining higher level circuits in hypothalamus as well as downstream pathways impacting on ANS and hypothalamic-pituitary axes in acute illness. Very exciting, careful and impactful work. I recommend its publication in Nature

Referee #2 (Remarks to the Author):

This is an elegant study which shows (surprisingly) that local PGE2 production following murine influenza signals through EP3 prostaglandin receptors in a small number of petrosal GABRA1 neurons to activate sickness behavior. An additional novel finding is that attenuation of sickness behavior, which has widely been viewed as a protective response, actually improves survival. The experiments supporting these conclusions are comprehensive, including the generation of multiple specific knock out models, supplemented by the use of prostaglandin receptor antagonists/antagonists, neuronal tracing, and targeted neuronal ablation.

Minor comments:

1. The effects of petrosal GABRA1 neuronal silencing via COX inhibitors, EP3 antagonist, and Ptger3 K/O appear only to be transient, with some observational variables (e.g., food and water intake) continuing to progressively worsen even as the untreated infected controls have begun to recover. It would be interesting to know the time dependent PGE2 systemic concentrations to help interpret this. Ibuprofen was given in the drinking water and the reduced consumption could conceivably lead to dosing that was suboptimal, explaining the late small rise in PGE2 as shown in Ext data Fig 1b. This rise mirrors in an inverse manner the behavioral variables. As aspirin was given parenterally and therefore would be dosed consistently, it would be interesting to know whether a similar late bump in PGE2 is observed. As PGE2 production is preserved in the Ptger3 K/O experiment (and presumably also when ganglionic neurons are K/O) do the authors have any thoughts about the very different response patterns in figure 2 b versus c, as compared to figure 1? As the authors point out, these observations indicate that additional pathways for activating illness behavior are involved, but it would be helpful if some suggestions could be proposed in the present brief discussion as a foundation for future work.

2. The authors comment that there was no hyperthermic response to infection. As reported by Jhaveri et al (Brain Behav Immun. 2007; 21(7): 975–987), mice infected with influenza virus maintained at 22 or 26°C develop hypothermia. Were there any changes/differences noted in core temperature, e.g., hypothermia, in control versus lesioned animals? Hypothermia has been shown to be protective in the setting of cytokine storm. Information about generalized inflammatory mediators (e.g., interleukins) would be interesting as these also elicit sickness behaviors.

3. Are there any thoughts concerning why (presumably) GABA responsive neurons mediate these effects? Are they potentially under chronic inhibition normally?

4. Study has shown that pathogens (e.g., Salmonella) can manipulate sickness behavior via inhibition of IL-1b with a resultant reduction in anorexia, leading to increased survival but at the cost of increased pathogen transmission (Rao, Cell 168:503 (2017)). The authors speculate that the potential relief of sickness behavior by manipulating the petrosal GABRA1 neurons could be beneficial, but could it not also lead to increased pathogen transmission in a similar way?

5. There are some typos. E.g. pages 8 and 9 – Fig 2 c,d should be Fig 3 c, d;

6. On page 11, AAV-GFP should be mentioned in addition to AAV-AP, as that is used to map NTS in Fig 5a.

7. In discussion, on page 14, they mentioned blocking PGE2 production attenuated...they should correct it to blocking PGE2 receptors, as no change in PGE2 production is seen (Ext Data Fig. 8).

Referee #3 (Remarks to the Author):

The manuscript titled “An airway-to-brain sensory pathway mediates influenza-induced sickness” describes a small population of vagal sensory neurons expressing Gabrar1 and the EP3 receptor that are involved in sickness-related behaviours resulting from an influenza infection. This is a good quality study from a lab that has previously published using these techniques and has a strong history working with vagal sensory neurobiology. The authors make excellent use of an existing data set to demonstrate distinct phenotypes of neurons that are differentially involved and establish one nodose subset that expresses Gabrar1. It was satisfying to see that an ‘older’ style method (glossopharyngeal nerve cut) worked in their favour and produced the same phenotype as the ‘newer’ methods such as transgenic animals and viral vector technology. The quality of the work is high but there are a few concerns that I have raised below.

Major concerns

1. We don't have any idea what any of these interventions are doing to the actual infection itself. It could be that infection severity is being modulated and what the authors are presenting is a secondary effect resulting from a decrease in infection severity. We know that sensory neurons are important for modulating and regulating immune responses and the authors even cite this literature themselves (Baral et al., 2018). At the very minimum, measurements of lung viral titers, lung histopathology, lung cytokines such as TNF α , IL6, IFN gamma etc and even serum cytokine levels need to be performed ideally as a time course across the course of infection to see what the infection itself is doing.

2. It appears what the authors are studying is an upper respiratory tract infection (URTI). Influenza infection with PR8 in mice is classically considered a lower respiratory tract infection (LRTI). Yes, there will be an initial URTI but in the latter phases it progresses to a LRTI. It appears at the later time points the phenotype involving the Phox2b and Gabrar1 neurons is lost indicative of their murine model of infection transitioning to the more classical LRTI. It would be more impactful if the authors could somehow demonstrate this, i.e. the sensory neurons are always playing a role but our model of infection is changing. The latter part of the model is different to the earlier part of the model. The authors need to consider, in addition to what has been presented, performing similar experiments but with a virus that is deemed to only cause a URTI in mice. It would also be beneficial to see how IAV progresses from the URT to the LRT across the time course of their studies. This could be done with standard immunohistochemistry techniques.

The true sickness behaviours related to IAV infections don't start until you get that classic LRTI and by that time point (around 6-10 days) the authors show that they appear to be losing their initial phenotype. The parameters measured such as body weight, food intake, water intake and motility across all of their interventions demonstrate that the experimental mice, around days 6-10, end up appearing similar to their control counterparts.

Perhaps the authors should consider changing this around to describe what drives morbidity in URTI. They demonstrate this URTI with their IAV model but I would also greatly consider using a virus that causes a classic URTI as a comparison. It is a little misleading to suggest that this glossopharyngeal Gabrar1 pathway regulates IAV-induced sickness behaviour as it only seems to in the early phases when the infection is considered an URTI. The bulk of IAV-induced sickness behaviour happens when the infection is considered a LRTI when things such as cytokine storms and hospitalisations are occurring.

3. The authors show that plasma PGE2 levels increase during infection (there is a systemic

availability of PGE2) yet the authors show that cutting the glossopharyngeal nerves produced a similar phenotype to what knocking-out or lesioning the neurons in the vagal ganglia did. This is a surprising result when all a nerve resection does is remove the peripheral terminals thereby not allowing those neurons to sense what is going on out in the periphery e.g. URT. However, the cell bodies are still intact which would make you believe that they can likely sense PGE2 systemically or, as we are aware, PGE2 readily enters the CNS through the BBB so the intact central terminals of these neurons could likely sense PGE2 via that route. So, it appears these neurons could still be activatable by PGE2 either systemically or centrally. Perhaps the plasma levels are not high enough to activate the neurons but if that is the case how then do you interpret the hypothalamic data?

Minor concerns

1. With the nestin knockout model, yes this will likely knockout neurons in the hypothalamus but how can the authors be sure that they aren't knocking out another group of neurons in another brain region that may have opposing effects to what the hypothalamus does? Perhaps a more specific approach such as local knockout approach is needed to truly assess the involvement of the hypothalamus.

2. The authors state that "We note that fever responses to influenza infection were not observed, perhaps because the ambient animal housing temperature was below their thermoneutral zone". Following infection with influenza mice do not typically display a febrile response, rather they undergo a hypothermic response. This has been reported widely by many researchers who use murine influenza models (Leyva-Grado et al., 2010; Dybing et al., 2000; Yang and Evans 1961; Bouvier and Lowen, 2010). Mouse can mount a febrile response but typically influenza does not do this. Ferrets seem to be the only animal model of influenza infection that display the fever response seen in humans. Did the authors measure body temperature?

3. Note that with animals that are housed alone this represents a potential stressor which could also impact on disease severity. With the measurements been made this is obviously something that can be overcome but perhaps something to think about.

The authors use the appropriate statistics and experimental controls in order to interpret their data. References are appropriate and up-to-date.

Author Rebuttals to Initial Comments:

Referees' comments:

Referee #1 (Remarks to the Author):

Sickness behaviors is a stereotypic response to acute infections and its role in host defense has been a subject of fascination but relatively little understanding of the underlying mechanisms. Here, the authors used state of the art neuroscience approaches to identify neuronal pathways responsible for the sickness behaviors induced by influenza infection. The study dramatically clarifies the contribution of central and peripheral neurons and identifies a small population of glossopharyngeal sensory neurons that are responsible for sickness behaviors in flu infection. These neurons express EP3 receptor, which is essential for orchestrating the behavioral response and change in homeostatic set points during illness. The study is very rigorous and conclusions are well supported and definitive. One interesting implication of the study is that different EP3 expressing sensory neurons that monitor different organ systems may be responsible for induction of sickness behaviors. This is important because it suggests that perhaps not all sickness behaviors are the same, which would certainly make physiological sense. It is also interesting that blocking sickness behaviors actually promoted survival if flu infections. There are maybe multiple possible explanations to this, including aspects of SPF conditions of the vivarium, or other ecological or evolutionary reasons that would be interesting to explore in the future.

In conclusion, this study is really a milestone in understanding sickness behaviors. It will provide the impetus for further studies examining higher level circuits in hypothalamus as well as downstream pathways impacting on ANS and hypothalamic-pituitary axes in acute illness. Very exciting, careful and impactful work. I recommend its publication in Nature

We thank the reviewer for these supportive comments.

Referee #2 (Remarks to the Author):

This is an elegant study which shows (surprisingly) that local PGE2 production following murine influenza signals through EP3 prostaglandin receptors in a small number of petrosal GABRA1 neurons to activate sickness behavior. An additional novel finding is that attenuation of sickness behavior, which has widely been viewed as a protective response, actually improves survival. The experiments supporting these conclusions are comprehensive, including the generation of multiple specific knock out models, supplemented by the use of prostaglandin receptor antagonists/antagonists, neuronal tracing, and targeted neuronal ablation.

We thank the reviewer for these supportive comments too.

Minor comments:

1. The effects of petrosal GABRA1 neuronal silencing via COX inhibitors, EP3 antagonist, and Ptger3 K/O appear only to be transient, with some observational variables (e.g., food and water intake) continuing to progressively worsen even as the untreated infected controls have begun to recover. It would be interesting to know the time dependent PGE2 systemic concentrations to help interpret this. Ibuprofen was given in the drinking water and the reduced consumption could conceivably lead to dosing that was suboptimal, explaining the late small rise in PGE2 as shown in Ext data Fig 1b. This rise mirrors in an inverse manner the behavioral variables. As aspirin was given parenterally and therefore would be dosed consistently, it would be interesting to know whether a similar late bump in PGE2 is observed. As PGE2 production is preserved in the Ptger3 K/O experiment (and presumably also when ganglionic neurons are K/O) do the authors have any thoughts about the very different response patterns in figure 2 b versus c, as compared to figure 1? As the authors point out, these observations indicate that additional pathways for activating illness behavior are involved, but it would be helpful if some suggestions could be proposed in the present brief discussion as a foundation for future work.

This is a terrific question, and we performed several new experiments that provide additional insight. First, we performed a similar time course involving aspirin as suggested. Aspirin and ibuprofen similarly attenuate flu-induced sickness behavior, and the dosage of aspirin utilized resulted in complete suppression of PGE2 production at all measured time points (Extended Data Fig. 1b). Thus, the same behavioral phenotype of attenuated flu-induced sickness was observed following (1) blockade of PGE2 production, (2) knockout of EP3 receptor in petrosal GABRA1 neurons, (3) ablation of petrosal GABRA1 neurons, and (4) glossopharyngeal nerve transection. Lower levels of residual sickness behavior are observed at later time points following all of these manipulations, indicating that residual sickness is mediated by a second neuronal pathway that is independent of PGE2, EP3, and glossopharyngeal sensory neurons. To explore this idea further, we additionally measured viral transcript levels in upper and lower airways, and observed that the second phase of sickness coincides with increased viral transcript levels in the lungs, which are primarily innervated by vagal rather than glossopharyngeal sensory neurons. This second pathway will be the subject of future studies, and it would be exciting to identify inhibitors which could be used in conjunction with NSAIDs to abolish both phases of flu-induced sickness behavior. As the reviewer suggests, we expand discussion of this point in the context of newly obtained data. We note that Figures 2b and 2c appeared different because monitoring was performed for a shorter duration in 2c. We have now repeated the experiment in Figure 2c and monitored animal behavior for 20 days, and found a general similarity of response patterns and recovery kinetics between 1, 2b, and 2c. Thank you for these suggestions.

2. The authors comment that there was no hyperthermic response to infection. As reported by Jhaveri et al (Brain Behav Immun. 2007; 21(7): 975–987), mice infected with influenza virus maintained at 22 or 26°C develop hypothermia. Were there any changes/differences noted in core temperature, e.g., hypothermia, in control versus lesioned animals? Hypothermia has been shown to be protective in the setting of cytokine storm. Information about generalized inflammatory mediators (e.g., interleukins) would be interesting as these also elicit sickness behaviors.

We revisited the body temperature response to flu infection using two approaches. First, we performed more extensive rectal temperature measurements, and second, we measured core body temperature using radiotelemetry with abdominally implanted telemetric transmitters. In both data sets, we now observed a clear hypothermic response to flu infection, and moreover, flu-induced hypothermia was attenuated in *Gabra1-ires-Cre; flox-Ptger3* mice. These new findings are presented in Extended Data 3a. We also performed new experiments to measure the levels of proinflammatory cytokines in control and *Gabra1-ires-Cre; flox-Ptger3* mice across a series of time points after flu infection. In control mice, we observed that BALF levels of several cytokines including IFN γ , TNF α , and IL-6 peaked in the lung five days after infection. Interestingly, in *Gabra1-ires-Cre; flox-Ptger3* mice, we observed that both lung viral transcripts and cytokine levels were lower and delayed. These new data are presented in Extended Data Fig. 9. Together, these findings indicate that EP3 receptor manipulations in peripheral sensory neurons not only impacts sickness behavior, but also impacts the hypothermia response and the immune response to infection.

3. Are there any thoughts concerning why (presumably) GABA responsive neurons mediate these effects? Are they potentially under chronic inhibition normally?

We performed additional experiments where flu-induced sickness behavior was monitored with or without daily administration of GABA (20 mg/kg, IP) in *flox-Ptger3* and *Gabra1-ires-Cre; flox-Ptger3* mice. Under these conditions, GABA administration did not impact the extent of sickness behavior in either of these cohorts (see Figure below). We do not understand the signaling role for GABRA1, if any, in this population of petrosal neurons, but nevertheless GABRA1 provides a terrific marker for the relevant neuronal subpopulation.

4. Study has shown that pathogens (e.g., Salmonella) can manipulate sickness behavior via inhibition of IL-1b with a resultant reduction in anorexia, leading to increased survival but at the cost of increased pathogen transmission (Rao, Cell 168:503 (2017)). The authors speculate that the potential relief of sickness behavior by manipulating the petrosal GABRA1 neurons could be beneficial, but could it not also lead to increased pathogen transmission in a similar way?

We completely agree with the reviewer, and have a sentence in the discussion about this possibility: "sickness behavior may provide a separate population-level benefit as sick animals seek isolation and thus may limit pathogen transmission to kin". We added the listed citation in support of that claim. We performed new experiments to monitor pathogen transmission by co-housing flu-infected animals of various genotypes with uninfected animals. However, we observed at best minor transmission in control animals that was only detected by measuring viral transcript levels, did not result in sickness behavior in co-housed animals, and was not enhanced by neuronal manipulations. We suspect a lack of efficient transmission in these experiments could very well be due to technical limitations associated with the flu strain used which is well known not to be highly transmissible, and/or the use of mice which do not cough. Furthermore, delineating effects of neuronal manipulations may require a customized (IACUC approved) co-housing paradigm that allows for more efficient social isolation of sick animals. For these reasons, we raise this idea only as a possibility in the discussion, but could delete the statement if the reviewer prefers.

5. There are some typos. E.g. pages 8 and 9 – Fig 2 c,d should be Fig 3 c, d;

Thank you- we made these corrections.

6. On page 11, AAV-GFP should be mentioned in addition to AAV-AP, as that is used to map NTS in Fig 5a.

AAV-GFP was only described in the figure legend, so we added description of AAV-GFP in the text as suggested.

7. In discussion, on page 14, they mentioned blocking PGE2 production attenuated...they should correct it to blocking PGE2 receptors, as no change in PGE2 production is seen (Ext Data Fig. 8).

We have extensively edited this paragraph in light of new data. The introductory sentence now reads: "Influenza infection-induced sickness behavior was attenuated, but not eliminated, following NSAID treatment, targeted EP3 receptor knockout, targeted neuronal ablation, and glossopharyngeal nerve transection, suggesting other routes to sickness..."

Referee #3 (Remarks to the Author):

The manuscript titled "An airway-to-brain sensory pathway mediates influenza-induced sickness" describes a small population of vagal sensory neurons expressing Gabrar1 and the EP3 receptor that are involved in sickness-related behaviours resulting from an influenza infection. This is a good quality study from a lab that has previously published using these techniques and has a strong history working with vagal sensory neurobiology. The authors make excellent use of an existing data set to demonstrate distinct phenotypes of neurons that are differentially involved and establish one nodose subset that expresses Gabrar1. It was satisfying to see that an 'older' style method (glossopharyngeal nerve cut)

worked in their favour and produced the same phenotype as the 'newer' methods such as transgenic animals and viral vector technology. The quality of the work is high but there are a few concerns that I have raised below.

We thank the reviewer for their supportive comments on the work.

Major concerns

1. We don't have any idea what any of these interventions are doing to the actual infection itself. It could be that infection severity is being modulated and what the authors are presenting is a secondary effect resulting from a decrease in infection severity. We know that sensory neurons are important for modulating and regulating immune responses and the authors even cite this literature themselves (Baral et al., 2018). At the very minimum, measurements of lung viral titers, lung histopathology, lung cytokines such as TNF α , IL6, IFN gamma etc and even serum cytokine levels need to be performed ideally as a time course across the course of infection to see what the infection itself is doing.

We performed new experiments to measure levels of viral transcript to infer viral load, as well as proinflammatory cytokines in control and *Gabra1-ires-Cre; flox-Ptger3* mice across a series of time points after flu infection. We measured transcript levels of the influenza A virus nucleoprotein gene as has been done previously (for example Yageta et al J Virol 2011, Wu et al Plos One 2012, Bao et al Front in Micro 2020, Kim et al Cell Mol Immunol 2022), and note that plaque assays for PR8 are challenging for technical reasons. In control mice, we observed that viral transcripts peaked in the lung five days after infection, and that BALF levels of several cytokines including IFN γ , TNF α , and IL-6 displayed a similar time course, also peaking five days after infection. Interestingly, in *Gabra1-ires-Cre; flox-Ptger3* mice, we observed that both lung viral transcripts and cytokine levels were lower and delayed. These new findings are presented in Extended Data Fig. 9. As will be discussed in the response to the second question below, the delayed time course matches the kinetics of viral transition from the upper to lower respiratory tract, which has exciting implications. Furthermore, these findings indicate that EP3 receptor manipulations in peripheral sensory neurons not only impacts an early wave of sickness behavior, but also impacts the immune response and the extent and timing of upper to lower respiratory tract transition.

A parsimonious interpretation of these findings is that petrosal GABRA1 neurons, upon detecting PGE2, engage neural circuits that evoke coordinated responses that include sickness behaviors as well as motor reflexes that impact immune function. It is also exciting to consider that changes in feeding behavior may secondarily impact immune function (as supported by Wang Cell 2016, Rao Cell 2017) and/or that petrosal GABRA1 neurons increase levels of other cytokines, which can also potentially elicit neuronal feedback and enhance sickness behavior. It is important to emphasize that each of these effects would critically depend on EP3 receptor in petrosal GABRA1 neurons, which clearly serves an essential role and first relays the presence of an upper respiratory infection to the brain, ultimately impacting sickness behavior. We add discussion of these points in the text. Together, our studies reveal the key neuronal site of PGE2 (and NSAID) action in neuro-immune crosstalk during influenza infection, which we believe is fundamental.

2. It appears what the authors are studying is an upper respiratory tract infection (URTI). Influenza infection with PR8 in mice is classically considered a lower respiratory tract infection (LRTI). Yes, there will be an initial URTI but in the latter phases it progresses to a LRTI. It appears at the later time points the phenotype involving the Phox2b and Gabrar1 neurons is lost indicative of their murine model of infection transitioning to the more classical LRTI. It would be more impactful if the authors could

somehow demonstrate this, i.e. the sensory neurons are always playing a role but our model of infection is changing. The latter part of the model is different to the earlier part of the model. The authors need to consider, in addition to what has been presented, performing similar experiments but with a virus that is deemed to only cause a URTI in mice. It would also be beneficial to see how IAV progresses from the URT to the LRT across the time course of their studies.

This could be done with standard immunohistochemistry techniques.

The true sickness behaviours related to IAV infections don't start until you get that classic LRTI and by that time point (around 6-10 days) the authors show that they appear to be losing their initial phenotype. The parameters measured such as body weight, food intake, water intake and motility across all of their interventions demonstrate that the experimental mice, around days 6-10, end up appearing similar to their control counterparts.

Perhaps the authors should consider changing this around to describe what drives morbidity in URTI. They demonstrate this URTI with their IAV model but I would also greatly consider using a virus that causes a classic URTI as a comparison. It is a little misleading to suggest that this glossopharyngeal GABRA1 pathway regulates IAV-induced sickness behaviour as it only seems to in the early phases when the infection is considered an URTI. The bulk of IAV-induced sickness behaviour happens when the infection is considered a LRTI when things such as cytokine storms and hospitalisations are occurring.

This is a really terrific question, and we have done additional experiments that provided new insights and significantly strengthened the paper. As the reviewer suggests, we measured the transition of infection from a URTI to a LRTI by measuring viral transcript levels in the upper and lower airways across a series of time points. In control mice, we observed that viral transcript levels in the upper airways peaked at day 3 while viral transcript levels in the lung peaked at day 5. In *Gabra1-ires-Cre; flox-Ptger3* mice, we observed a partial reduction of viral transcript levels in the upper airways, and then decreased and delayed viral transcript levels in the lungs that were measurable for a longer duration (Extended Data Fig. 9). What is particularly exciting is that the delayed viral transcript levels in the lungs precisely matches the kinetics of the attenuated sickness behavior observed in mice with deficient PGE2 signaling, through either 1) NSAID treatment, 2) targeted EP3 receptor knockout, 3) targeted neuronal ablation, or 4) glossopharyngeal nerve transection. As the reviewer predicts, these findings indicate that there are likely two phases of flu-induced sickness behavior. The first phase occurs when the virus is most prevalent in the upper respiratory tract, and sickness is primarily mediated by PGE2-detecting glossopharyngeal sensory neurons marked by GABRA1 which project to the nasopharynx. The second phase of sickness occurs when the virus is most prevalent in the lower respiratory tract, and we note that the lung is primarily innervated by vagal rather than glossopharyngeal sensory neurons. Our data raise the possibility that this phase of sickness involves a second neuronal pathway that is independent of PGE2, EP3, and glossopharyngeal sensory neurons. This second pathway will be the subject of future studies, and it would be exciting for example to identify inhibitors which could be used in conjunction with NSAIDs to abolish both phases of sickness behavior. We have edited the abstract and discussion to clarify that observed effects are 'during early-stage infection'. Our animal protocol is not approved to use any URTI-restricted viruses, but we are excited that neuronal manipulations produced such a clear phenotype in the more complex flu model, and could help disentangle contributions to sickness behavior from infection of the upper and lower respiratory tracts. We think these new findings are exciting and clarifying for the study, and have revised the text accordingly. Thank you for raising this point.

3. The authors show that plasma PGE2 levels increase during infection (there is a systemic availability of PGE2) yet the authors show that cutting the glossopharyngeal nerves produced a similar phenotype to what knocking-out or lesioning the neurons in the vagal ganglia did. This is a surprising result when all a

nerve resection does is remove the peripheral terminals thereby not allowing those neurons to sense what is going on out in the periphery e.g. URT. However, the cell bodies are still intact which would make you believe that they can likely sense PGE2 systemically or, as we are aware, PGE2 readily enters the CNS through the BBB so the intact central terminals of these neurons could likely sense PGE2 via that route. So, it appears these neurons could still be activatable by PGE2 either systemically or centrally. Perhaps the plasma levels are not high enough to activate the neurons but if that is the case how then do you interpret the hypothalamic data?

Thank you for raising this discussion. Our data clearly indicate that both GABRA1 neurons and their peripheral axons are essential for detecting tissue-localized PGE2 in the context of flu infection in the upper respiratory tract. Targeted neuronal manipulations argue strongly against alternative models that flu-induced PGE2 acts systemically in the context of our experiments- either on the soma of peripheral sensory neurons or directly in the hypothalamus. Other sites for neuronal detection of EP3 may be relevant in other infection models, but our data indicate that alternate pathways which persist after loss of petrosal GABRA1 axons are not sufficient to mediate normal sickness responses to early-stage flu infection. The principally cited prior study that implicated direct PGE2 detection in the hypothalamus was based on LPS-induced fever rather than a naturalistic airway infection. IP injection of high levels of LPS presumably triggers a distinct bacteria-related and perhaps supraphysiological response that is not relevant for flu infection, and furthermore, that study did not look at the numerous behavioral measures that we report here. For these reasons, we considered it critical to look at sickness responses to a naturalistic infection. We observed that ablation of central EP3 receptors (including in the hypothalamus) had no impact on flu-induced sickness, while instead eliminating peripheral EP3 receptors in petrosal GABRA1 neurons through numerous approaches (including various targeted EP3 knockouts, neuron ablations and glossopharyngeal nerve transection) evoked a strong phenotype. The requirement for peripheral axons supports our anatomical observations that the sensory axons of GABRA1 neurons project near sites of PGE2 production in the airways rather than being anatomically positioned to detect PGE2 in plasma. While it is not readily possible to measure local PGE2 levels precisely at the peripheral nerve terminal in the nasopharynx, they presumably precede and exceed systemic levels in circulation. Nerve transection is a classical approach to eliminate the responses of vagal, glossopharyngeal, and other peripheral sensory neurons, as most, if not all, known vagal/glossopharyngeal responses are critically dependent on the sensory axon. Together, these findings indicate a parsimonious model whereby airway-innervating neurons provide a fast and robust conduit for information transfer from the airways to the brain. It is for these very reasons that we consider our findings revealing a critical role for peripheral PGE2 detection in sickness to be exciting and novel.

Minor concerns

1. With the nestin knockout model, yes this will likely knockout neurons in the hypothalamus but how can the authors be sure that they aren't knocking out another group of neurons in another brain region that may have opposing effects to what the hypothalamus does? Perhaps a more specific approach such as local knockout approach is needed to truly assess the involvement of the hypothalamus.

The reviewer is suggesting the possibility that PGE2 not only activates pro-sickness pathways in the hypothalamus and peripheral nerves, but also a third anti-sickness pathway additionally marked in Nestin-Cre mice. Since Flu induces sickness in control mice, pro-sickness pathways must be dominant over any hypothetical anti-sickness pathway. The main point of the manuscript is that loss of petrosal GABRA1 neurons eliminates the dominant pro-sickness pathway, which is clearly supported by our numerous neuronal manipulations. If Nestin-Cre mice were the only manipulation we performed, we agree that we would not be able to draw strong conclusions, and would worry about alternative

counterregulatory and/or redundant PGE2-detection pathways. However, we additionally performed numerous manipulations in peripheral neurons which revealed a strong phenotype similar in magnitude to ibuprofen, and found no evidence suggesting another key PGE2 detection pathway in the context of the flu model. We note that AAV-Cre mediated knockout requires high efficiency; the hypothalamus presents a difficult (and potentially insurmountable) technical challenge for AAV-mediated knockout as the hypothalamus is a larger and less contained structure than peripheral ganglia. We also added a new experiment (Extended Data Fig. 6c) to measure EP3 levels in cell-specific knockout mice which help verify the selectivity of our manipulations. We agree with the reviewer that alternate PGE2 detection pathways may be engaged during other infection paradigms, and discuss this in the manuscript.

2. The authors state that “We note that fever responses to influenza infection were not observed, perhaps because the ambient animal housing temperature was below their thermoneutral zone”. Following infection with influenza mice do not typically display a febrile response, rather they undergo a hypothermic response. This has been reported widely by many researchers who use murine influenza models (Leyva-Grado et al., 2010; Dybing et al., 2000; Yang and Evans 1961; Bouvier and Lowen, 2010). Mice can mount a febrile response but typically influenza does not do this. Ferrets seem to be the only animal model of influenza infection that display the fever response seen in humans. Did the authors measure body temperature?

We revisited the body temperature response to flu infection using two approaches. First, we performed more extensive rectal temperature measurements, and second, we measured core body temperature using radiotelemetry with abdominally implanted telemetric transmitters every 15 minutes post-infection. In both data sets, we now observed a clear hypothermic response to flu infection, and moreover, flu-induced hypothermia was attenuated in *Gabra1-ires-Cre; flox-Ptger3* mice. These new findings are presented in Extended Data 3a.

3. Note that with animals that are housed alone this represents a potential stressor which could also impact on disease severity. With the measurements been made this is obviously something that can be overcome but perhaps something to think about.

We performed additional experiments where animals were group-housed after flu infection. The extent of sickness behavior was similar in singly housed and group-housed animals, and furthermore, a similar decrease in sickness responses was observed in singly housed and group-housed *Gabra1-ires-Cre; flox-Ptger3* mice. Please see the figure below for more information.

The authors use the appropriate statistics and experimental controls in order to interpret their data. References are appropriate and up-to-date.

Thank you for noting this.

Reviewer Reports on the First Revision:

Referees' comments:

Referee #1 (Remarks to the Author):

The manuscript is further improved during the revision and the additional data provide important new insights into physiological control of sickness behavior. I don't have any further comments and think the study is a valuable contribution to an important and still understudied field and would be of broad interest.

Referee #2 (Remarks to the Author):

The results show a population of glossopharyngeal sensory neurons (petrosal GABRA1 neurons) mediate influenza-induced sickness behavior in mice. Ablating petrosal GABRA1 neurons, or targeted knockout of PGE2 receptor 3 (EP3) in these neurons, eliminates influenza-induced decreases in food intake, water intake, and mobility during early-stage infection, and improves survival. The authors conclude that this is "a primary airway-to-brain sensory pathway that detects locally produced prostaglandins and mediates systemic sickness responses to respiratory virus infection."

The results are highly original, significant, and represent a major advance to the field.

The use of statistics is appropriate, and the results support the conclusions.

The authors have responded extensively to prior review, and their new results and responses have improved the paper, strengthened the original results, and bolstered the original conclusions.

The manuscript is clear and concise. The prose delivers the conclusions as a well-supported argument in the style of other classic scientific articles.

Author Rebuttals to First Revision:

We thank the referees for the time spent working on the manuscript, and appreciate that they had no further queries.